# PhyloGFN: Phylogenetic inference with generative flow networks

**Mingyang Zhou**[*]
McGill University

**Zichao Yan**[*]
McGill University
Université de Montréal, Mila

**Elliot Layne**
McGill University, Mila

**Nikolay Malkin**
Université de Montréal, Mila

**Dinghuai Zhang**
Université de Montréal, Mila

**Moksh Jain**
Université de Montréal, Mila

**Mathieu Blanchette**
McGill University, Mila

**Yoshua Bengio**[†]
Université de Montréal, Mila

```
{ming.zhou,zichao.yan,elliot.layne}@mail.mcgill.ca, blanchem@cs.mcgill.ca
{nikolay.malkin,dinghuai.zhang,moksh.jain,yoshua.bengio}@mila.quebec
```

## Abstract

Phylogenetics is a branch of computational biology that studies the evolutionary relationships among biological entities. Its long history and numerous applications notwithstanding, inference of phylogenetic trees from sequence data remains challenging: the extremely large tree space poses a significant obstacle for the current combinatorial and probabilistic techniques. In this paper, we adopt the framework of generative flow networks (GFlowNets) to tackle two core problems in phylogenetics: parsimony-based and Bayesian phylogenetic inference. Because GFlowNets are well-suited for sampling complex combinatorial structures, they are a natural choice for exploring and sampling from the multimodal posterior distribution over tree topologies and evolutionary distances. We demonstrate that our amortized posterior sampler, PhyloGFN, produces diverse and high-quality evolutionary hypotheses on real benchmark datasets. PhyloGFN is competitive with prior works in marginal likelihood estimation and achieves a closer fit to the target distribution than state-of-the-art variational inference methods.

## 1 Introduction

Phylogenetic inference has long been a central problem in the field of computational biology. Accurate phylogenetic inference is critical for a number of important biological analyses, such as understanding the development of antibiotic resistance (Aminov & Mackie, 2007; Ranjbar et al., 2020; Layne et al., 2020), assessing the risk of invasive species (Hamelin et al., 2022; Dort et al., 2023), and many others. Accurate phylogenetic trees can also be used to improve downstream computational analyses, such as multiple genome alignment (Blanchette et al., 2004), ancestral sequence reconstruction (Ma et al., 2006), protein structure and function annotation (Celniker et al., 2013).

Despite its strong medical relevance and wide applications in life science, phylogenetic inference has remained a standing challenge, in part due to the high complexity of tree space — for $n$ species, $(2n - 5)!!$ unique unrooted bifurcating tree topologies exist. This poses a common obstacle to all branches of phylogenetic inference; both maximum-likelihood and maximum-parsimony tree reconstruction are NP-hard problems (Day, 1987; Chor & Tuller, 2005). Under the Bayesian formulation of phylogenetics, the inference problem is further compounded by the inclusion of continuous variables that capture the level of sequence divergence along each branch of the tree.

One line of prior work considers Markov chain Monte Carlo (MCMC)-based approaches, such as MrBayes (Ronquist et al., 2012). These approaches have been successfully applied to Bayesian phylogenetic inference. However, a known limitation of MCMC is scalability to high-dimensional

---

[*]Equal contribution. [†]CIFAR Senior Fellow.

distributions with multiple separated modes (Tjelmeland & Hegstad, 2001), which arise in larger phylogenetic datasets. Recently, variational inference (VI)-based approaches have emerged. Among these methods, some model only a limited portion of the space of tree topologies, while others are weaker in marginal likelihood estimation due to simplifying assumptions. In parsimony analysis, state-of-the-art methods such as PAUP* (Swofford, 1998) have extensively relied on heuristic search algorithms that are efficient but lack theoretical foundations and guarantees.

Coming from the intersection of variational inference and reinforcement learning is the class of models known as generative flow networks (GFlowNets; Bengio et al., 2021). The flexibility afforded by GFlowNets to learn sequential samplers for distributions over compositional objects makes them a promising candidate for performing inference over the posterior space of phylogenetic tree topologies and evolutionary distances.

In this work, we propose **PhyloGFN**, the first adaptation of GFlowNets to the task of Bayesian and parsimony-based phylogenetic inference. Our contributions are as follows:

(1) We design an acyclic Markov decision process (MDP) with fully customizable reward functions, by which our PhyloGFN can be trained to construct phylogenetic trees in a bottom-up fashion.
(2) PhyloGFN leverages a novel tree representation inspired by Fitch and Felsenstein's algorithms to represent rooted trees without introducing additional learnable parameters to the model. PhyloGFN is also coupled with simple yet effective training techniques such as using mixed on-policy and dithered-policy rollouts, replay buffers and cascading temperature-annealing.
(3) PhyloGFN has the capacity to explore and sample from the entire phylogenetic tree space, achieving a balance between exploration in this vast space and high-fidelity modeling of the modes. While PhyloGFN performs on par with the state-of-the-art MCMC- and VI-based methods in the summary metric of marginal log-likelihood, it substantially outperforms these approaches in terms of its ability to estimate the posterior probability of suboptimal trees.

## 2 RELATED WORK

Markov chain Monte Carlo (MCMC)-based algorithms are commonly employed for Bayesian phylogenetics, with notable examples including MrBayes and RevBayes (Ronquist et al., 2012; Höhna et al., 2016), which are considered state-of-the-art in the field. Amortized variational inference (VI) is an alternative approach that parametrically estimates the posterior distribution. VBPI-GNN (Zhang, 2023) employs subsplit Bayesian networks (SBN) (Zhang & Matsen IV, 2018a) to model tree topology distributions and uses graph neural networks to learn tree topological embeddings (Zhang, 2023). While VBPI-GNN has obtained marginal log likelihood competitive with MrBayes in real datasets, it requires a pre-generated set of high-quality tree topologies to constrain its action space for tree construction, which ultimately limits its ability to model the entire tree space.

There exist other VI approaches that do not limit the space of trees. VaiPhy (Koptagel et al., 2022) approximates the posterior distribution in the augmented space of tree topologies, edge lengths, and ancestral sequences. Combined with combinatorial sequential Monte Carlo (CSMC; Moretti et al., 2021), the proposed method enables faster estimation of marginal likelihood. GeoPhy (Mimori & Hamada, 2023) models the tree topology distribution in continuous space by mapping continuous-valued coordinates to tree topologies, using the same technique as VBPI-GNN to model tree topological embeddings. While both methods model the entire tree topology space, their performance on marginal likelihood estimation underperforms the state of the art.

For the optimization problem underpinning maximum parsimony inference, PAUP* is one of the most commonly used programs (Swofford, 1998); it features several fast, greedy, and heuristic algorithms based on local branch-swapping operations such as tree bisection and reconnection.

GFlowNets are a family of methods for sampling discrete objects from multimodal distributions, such as molecules (Bengio et al., 2021) and biological sequences (Jain et al., 2022), and are used to solve discrete optimization tasks (Zhang et al., 2023a;b). With their theoretical foundations laid out in Bengio et al. (2023); Lahlou et al. (2023), and connections to variational inference established in Malkin et al. (2023), GFlowNets have been successfully applied to tackle complex Bayesian inference problems, such as inferring latent causal structures in gene regulatory networks (Deleu et al., 2022; 2023), and parse trees in hierarchical grammars (Hu et al., 2023).

## 3 BACKGROUND

### 3.1 PHYLOGENETIC INFERENCE

Here we introduce the problems of Bayesian and parsimony-based phylogenetic inference. A weighted phylogenetic tree is denoted by $(z, b)$, where $z$ represents the tree topology with its leaves labeled by observed sequences, and $b$ represents the branch lengths. The tree topology can be either a rooted binary tree or a bifurcating unrooted tree. For a tree topology $z$, let $L(z)$ denote the labeled sequence set and $E(z)$ the set of edges. For an edge $e \in E(z)$, let $b(e)$ denote its length. Let $Y = \{y_1, y_2 \ldots y_n\} \in \Sigma^{n \times m}$ be a set of $n$ observed sequences, each having $m$ characters from alphabet $\Sigma$, *e.g.*, $\{A, C, G, T\}$ for DNA sequences. We denote the $i^{\text{th}}$ site of all sequences by $Y^i = \{y_1[i], y_2[i] \ldots y_n[i]\}$. In this work, we make two assumptions that are common in the phylogenetic inference literature: (i) sites evolve independently; (ii) evolution follows a time-reversible substitution model. The latter implies that an unrooted tree has the same parsimony score or likelihood as its rooted versions, and thus the algorithms we introduce below (Fitch and Felsenstein) apply to unrooted trees as well.

#### 3.1.1 BAYESIAN INFERENCE

In Bayesian phylogenetic inference, we are interested in sampling from the posterior distribution over weighted phylogenetic trees $(z, b)$, formulated as:

$$P(z, b \mid Y) = \frac{P(z, b)P(Y \mid z, b)}{P(Y)}$$

where $P(Y \mid z, b)$ is the likelihood, $P(Y)$ is the intractable marginal, and $P(z, b)$ is the prior density over tree topology and branch lengths. Under the site independence assumption, the likelihood can be factorized as: $P(Y \mid z, b) = \prod_i P(Y^i \mid z, b)$, and each factor is obtained by marginalizing over all internal nodes $a_j$ and their possible character assignment:

$$P(y_1[i] \ldots y_n[i] \mid z, b) = \sum_{a_{n+1}^i, \ldots a_{2n-1}^i} P(a_{2n-1}) \prod_{j=n+1}^{2n-2} P(a_j^i | a_{\alpha(j)}^i, b(e_j)) \prod_{k=1}^{n} P(y_k[i] | a_{\alpha(k)}^i, b_z(e_k))$$

where $a_{n+1}^i, \ldots a_{2n-2}^i$ represent the internal node characters assigned to site $i$ and $\alpha(i)$ represent the parent of node $i$. $P(a_{2n-1})$ is a distribution at the root node, which is usually assumed to be uniform over the vocabulary, while the conditional probability $P(a_j^i | a_{\alpha(j)}^i, b(e_j))$ is defined by the substitution model (where $e_j$ is the edge linking $a_j$ to $\alpha(a_j)$).

**Felsenstein's algorithm** The likelihood of a given weighted phylogenetic tree can be calculated efficiently using Felsenstein's pruning algorithm (Felsenstein, 1973) in a bottom-up fashion through dynamic programming. Defining $L_u^i$ as the leaf sequence characters at site $i$ below the internal node $u$, and given its two child nodes $v$ and $w$, the conditional probability $P(L_u^i | a_u^i)$ can be obtained from $P(L_v^i | a_v^i)$ and $P(L_w^i | a_w^i)$:

$$P(L_u^i \mid a_u^i) = \sum_{a_v^i, a_w^i \in \Sigma} P(a_v^i \mid a_u^i, b(e_v)) P(L_v^i \mid a_v^i) P(a_w^i \mid a_u^i, b(e_w)) P(L_w^i \mid a_w^i). \tag{1}$$

The dynamic programming, or recursion, is essentially a post-order traversal of the tree and $P(L_u^i \mid a_u^i)$ is calculated at every internal node $u$, and we use one-hot encoding of the sequences to represent the conditional probabilities at the leaves. Finally, the conditional probability for each node $u$ at site $i$ is stored in a data structure $f_u^i \in [0, 1]^{|\Sigma|}$: $f_u^i[c] = P(L_u^i | a_u^i = c)$, and we call it the **Felsenstein feature** for node $u$. Note that the conditional probability at the root $P(Y^i | a_{2n-1}^i)$ is used to calculate the likelihood of the tree: $P(Y^i | z, b) = \sum_{a_{2n-1}^i \in \Sigma} P(a_{2n-1}^i) P(Y^i | a_{2n-1}^i)$.

#### 3.1.2 PARSIMONY ANALYSIS

The problem of finding the optimal tree topology under the maximum parsimony principle is commonly referred as the Large Parsimony problem, which is NP-hard. For a given tree topology $z$, the parsimony score is the minimum number of character changes between sequences across branches obtained by optimally assigning sequences to internal nodes. Let $M(z|Y)$ be the parsimony score of tree topology $z$ with leaf labels $Y$. Due to site independence, $M(z|Y) = \sum_i M(z|Y^i)$. The trees with the lowest parsimony score, or most parsimonious trees, are solutions to the Large Parsimony problem. Note that the Large Parsimony problem is a limiting case of the maximum likelihood tree inference problem, where branch lengths are constrained to be equal and infinitesimally short.

**Fitch algorithm** Given a rooted tree topology $z$, the Fitch algorithm assigns optimal sequences to internal nodes and computes the parsimony score in linear time. At each node $u$, the algorithm tracks the set of possible characters labeling for node $u$ that can yield a most parsimonious solution for the subtree rooted at $u$. This character set can be represented by a binary vector $\boldsymbol{f}_u^i \in \{0, 1\}^{|\Sigma|}$ for site $i$. We label this vector the **Fitch feature**. As in Felsenstein's algorithm, this vector is a one-hot encoding of the sequences at the leaves and is computed recursively for non-leaves. Specifically, given a rooted tree with root $u$ and two child trees with roots $v$ and $w$, $\boldsymbol{f}_u^i$ is calculated as:

$$\boldsymbol{f}_u^i = \begin{cases} \boldsymbol{f}_v^i \wedge \boldsymbol{f}_w^i & \text{if } \boldsymbol{f}_v^i \cdot \boldsymbol{f}_w^i \neq 0 \\ \boldsymbol{f}_v^i \vee \boldsymbol{f}_w^i & \text{otherwise} \end{cases},$$

where $\wedge$ and $\vee$ are element-wise conjunctions and disjunctions. The algorithm traverses the tree two times, first in post-order (bottom-up) to calculate the character set at each node, then in pre-order (top-down) to assign optimal sequences. The total number of character changes between these optimal sequences along the tree's edges is counted as the parsimony score.

## 3.2 GFLOWNETS

Generative flow networks (GFlowNets) are algorithms for learning generative models of complex distributions given by unnormalized density functions over structured spaces. Here, we give a concise summary of the the GFlowNet framework.

**Setting** A GFlowNet treats generation of objects $x$ lying in a sample space $\mathcal{X}$ as a sequential decision-making problem on an acyclic deterministic MDP with set of states $\mathcal{S} \supset \mathcal{X}$ and set of actions $\mathcal{A} \subseteq \mathcal{S} \times \mathcal{S}$. The MDP has a designated *initial state* $s_0$, which has no incoming actions, and a set of *terminal states* (those with no outgoing actions) that coincides with $\mathcal{X}$. Any $x \in \mathcal{X}$ can be reached from $s_0$ by a sequence of actions $s_0 \to s_1 \to \cdots \to s_n = x$ (with each $(s_i, s_{i+1}) \in \mathcal{A}$). Such sequences are called *complete trajectories*, and the set of all complete trajectories is denoted $\mathcal{T}$.

A *(forward) policy* $P_F$ is a collection of distributions $P_F(s' \mid s)$ over the children of each nonterminal state $s \in \mathcal{S} \setminus \mathcal{X}$. A policy induces a distribution over $\mathcal{T}$:

$$P_F(\tau = (s_0 \to s_1 \to \cdots \to s_n)) = \prod_{i=0}^{n-1} P_F(s_{i+1} \mid s_i).$$

A policy gives a way to sample objects in $\mathcal{X}$, by sampling a complete trajectory $\tau \sim P_F$ and returning its final state, inducing a marginal distribution $P_F^\top$ over $\mathcal{X}$; $P_F^\top(x)$ is the sum of $P_F(\tau)$ over all complete trajectories $\tau$ that end in $x$ (a possibly intractable sum).

The goal of GFlowNet training is to estimate a parametric policy $P_F(\cdot \mid \cdot; \theta)$ such that the induced $P_F^\top$ is proportional to a given *reward function* $R : \mathcal{X} \to \mathbb{R}_{>0}$, *i.e.*,

$$P_F^\top(x) = \frac{1}{Z} R(x) \quad \forall x \in \mathcal{X}, \tag{2}$$

where $Z = \sum_{x \in \mathcal{X}} R(x)$ is the unknown normalization constant (partition function).

**Trajectory balance objective** The direct optimization of $P_F$'s parameters $\theta$ is impossible since it involves an intractable sum over all complete trajectories. Instead, we leverage the trajectory balance (TB) training objective (Malkin et al., 2022), which introduces two auxiliary objects: an estimate $Z_\theta$ of the partition function and a *backward policy*. In our experiments, we fix the backward policy to uniform, which results in a simplified objective:

$$\mathcal{L}_{\text{TB}}(\tau) = \left( \log \frac{Z_\theta \prod_{i=0}^{n-1} P_F(s_{i+1} \mid s_i; \theta)}{R(x) P_B(\tau \mid x)} \right)^2, \quad P_B(\tau \mid x) := \prod_{i=1}^{n} \frac{1}{|\text{Pa}(s_i)|}, \tag{3}$$

where $\text{Pa}(s)$ denotes the set of parents of $s$. By the results of Malkin et al. (2022), there exists a unique policy $P_F$ and scalar $Z_\theta$ that simultaneously make $\mathcal{L}_{\text{TB}}(\tau) = 0$ for all $\tau \in \mathcal{T}$, and at this optimum, the policy $P_F$ satisfies (2) and $Z_\theta$ equals the true partition function $Z$. In practice, the policy $P_F(\cdot \mid s; \theta)$ is parameterized as a neural network that outputs the logits of actions $s \to s'$ given a representation of the state $s$ as input, while $Z_\theta$ is parameterized in the log domain.

$\mathcal{L}_{\text{TB}}(\tau)$ is minimized by gradient descent on trajectories $\tau$ chosen from a behaviour policy that can encourage exploration to accelerate mode discovery (see our behaviour policy choices in §4.3).

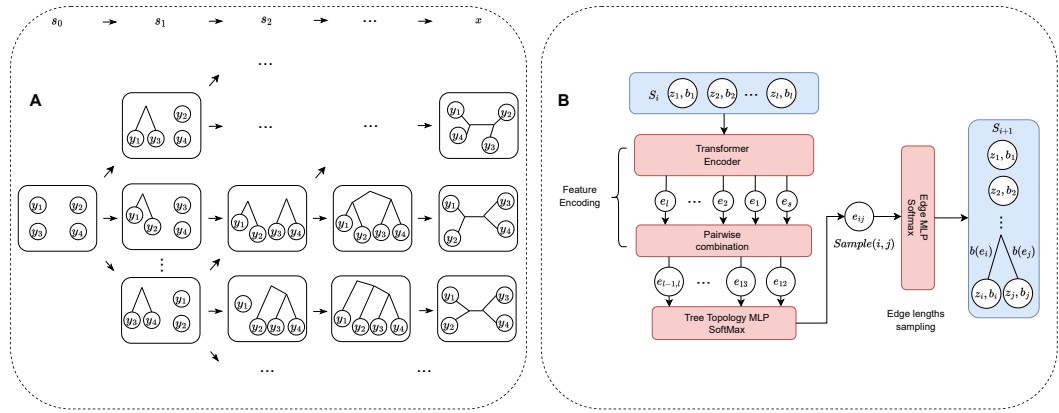

Figure 1: **Left:** PhyloGFN's state space on a four-sequence dataset. Initial state $s_0$ comprises leaf nodes. Successive steps merge pairs of trees until a single unrooted tree remains. **Right:** Policy model for PhyloGFN-Bayesian. Transformer encoder processes tree-level features $s_i = ((z_1, b_1) \dots (z_l, b_l))$. Pairwise features $e_{ij}$ are derived and used by MLPs to select tree pairs for merging and sample branch lengths.

## 4 PHYLOGENETIC INFERENCE WITH GFLOWNETS

### 4.1 GFLOWNETS FOR BAYESIAN PHYLOGENETIC INFERENCE

This section introduces PhyloGFN-Bayesian, our GFlowNet-based method for Bayesian phylogenetic inference. Given a set of observed sequences $Y$, PhyloGFN-Bayesian learns a sampler over the joint space of tree topologies and edge lengths $\mathcal{X} = \{(z, b)|L(z) = Y\}$ such that the sampling probability of $(z, b) \in \mathcal{X}$ approximates its posterior $P_F^\top(z, b) = P(z, b|Y)$.

We follow the same setup as Koptagel et al. (2022); Zhang (2023); Mimori & Hamada (2023): (i) uniform prior over tree topologies; (ii) decomposed prior $P(z, b) = P(z)P(b)$; (iii) exponential ($\lambda = 10$) prior over branch lengths; (iv) Jukes-Cantor substitution model.

**GFlowNet state and action space** The sequential procedure of constructing phylogenetic trees is illustrated in Fig. 1. The initial state $s_0$ is a set of $n$ rooted trees, each containing a single leaf node labeled with an observed sequence. Each action chooses a pair of trees and joins their roots by a common parent node, thus creating a new tree. The number of rooted trees in the set is reduced by 1 at every step, so after $n - 1$ steps a single rooted tree with $n$ leaves is left. To obtain an unrooted tree, we simply remove the root node.

Thus, a state $s$ consists of a set of disjoint rooted trees $s = ((z_1, b_1), \dots, (z_l, b_l))$, $l \leq n$ and $\bigcup_i L(z_i) = Y$. Given a nonterminal state with $l > 1$ trees, a transition action consists of two steps: (i) choosing a pair of trees to join out of the $\binom{l}{2}$ possible pairs; and (ii) generating branch lengths for the two introduced edges between the new root and its two children. The distribution over the pair of branch lengths is modeled jointly as a discrete distribution with fixed bin size. Following the initial submission of the paper, we conducted additional experiments by employing continuous distribution to model branch lengths. Further details can be found in §F of the appendix.

**Reward function** We define the reward function as the product of the likelihood and the edge length prior: $R(z, b) = P(Y|z, b)P(b)$, implicitly imposing a uniform prior over tree topologies. By training with this reward, PhyloGFN learns to approximate the posterior, since $P(z, b|Y) = R(z, b)\frac{P(z)}{P(Y)}$ and $P(z), P(Y)$ are both constant.

It is worth emphasizing that in our bottom-up construction of trees, the set of possible actions at the steps that select two trees to join by a new common root is never larger than $n^2$, even though the size of the space of all tree topologies – all of which can be reached by our sampler – is superexponential in $n$. This stands in contrast to the modeling choices of VBPI-GNN (Zhang, 2023), which constructs trees in a top-down fashion and limits the action space using a pre-generated set of trees, therefore also limiting the set of trees it can sample.

**State representation** To represent a rooted tree in a non-terminal state, we compute features for each site independently by taking advantage of the Felsenstein features (§3.1.1). Let $(z, b)$ be a weighted tree with root $u$ which has two children $v$ and $w$. Let $f_u^i, f_v^i, f_w^i \in [0, 1]^{|\Sigma|}$ be the Felsen-

stein feature on nodes $u, v, w$ at site $i$. The representation $\rho_u^i$ for site $i$ is computed as following:

$$\rho_u^i = f_u^i \prod_{e \in E(z)} P(b_z(e))^{\frac{1}{m}} \tag{4}$$

where $P(b_z(e)) = \prod_{e \in b_z} P(b(e))$ is the edge length prior. The tree-level feature is the concatenation of site-level features $\rho = [\rho^1 \dots \rho^m]$. A state $s = (z_1 \dots z_l)$, which is a collection of rooted trees, is represented by the set $\{\rho_1, \dots, \rho_l\}$.

**Representation power** Although the proposed feature representation $\rho$ does not capture all the information of tree structure and leaf sequences, we show that $\rho$ indeed contains sufficient information to express the optimal policy. The proposition below shows that given an optimal GFlowNet with uniform $P_B$, two states with identical features set share the same transition probabilities.

**Proposition 1.** *Let* $s_1 = \{(z_1, b_1), (z_2, b_2) \dots (z_l, b_l)\}$ *and* $s_2 = \{(z_1', b_1'), (z_2', b_2') \dots (z_l', b_l')\}$ *be two non-terminal states such that* $s_1 \neq s_2$ *but sharing the same features* $\rho_i = \rho_i'$. *Let* $\boldsymbol{a}$ *be any sequence of actions, which applied to* $s_1$ *and* $s_2$, *respectively, results in full weighted trees* $x = (z, b_z), x' = (z', b')$, *with two partial trajectories* $\tau = (s_1 \to \cdots \to x), \tau' = (s_2 \to \cdots \to x')$. *If* $P_F$ *is the policy of an optimal GFlowNet with uniform* $P_B$, *then* $P_F(\tau) = P_F(\tau')$.

All proofs are in Appendix A. The proposition shows that our proposed features have sufficient representation power for the PhyloGFN-Bayesian policy. Furthermore, Felsenstein features and edge length priors are used in calculating reward by Felsenstein's algorithm. Therefore, computing these features does not introduce any additional variables, and computation overhead is minimized.

### 4.2 GFLOWNETS FOR PARSIMONY ANALYSIS

This section introduces PhyloGFN-Parsimony, our GFlowNet-based method for parsimony analysis. We treat large parsimony analysis, a discrete optimization problem, as a sampling problem from the energy distribution $\exp\left(\frac{-M(z|Y)}{T}\right)$ defined over tree topologies. Here, $M(z|Y)$ is the parsimony score of $z$ and $T$ is a pre-defined temperature term to control the smoothness of distribution. With sufficiently small $T$, the most parsimonious trees dominate the energy distribution. To state our goals formally, given observed sequences $Y$, PhyloGFN-Parsimony learns a sampling policy $P_F$ over the space of tree topologies $\{z|L(z) = Y\}$ such that $P_F^\top(z) \propto e^{-\frac{M(z|Y)}{T}}$. As $T \to 0$, this target distribution approaches a uniform distribution over the set of tree topologies with minimum parsimony scores.

PhyloGFN-Parsimony can be seen as a reduced version of PhyloGFN-Bayesian. The tree shape generation procedure is the same as before, but we no longer generate branch lengths. The reward is defined as $R(z) = \exp\left(\frac{C - M(z|Y)}{T}\right)$, where $C$ is an extra hyperparameter introduced for stability to offset the typically large $M(z|Y)$ values. Note that $C$ can be absorbed into the partition function and has no influence on the reward distribution.

Similar to PhyloGFN-Bayesian, a state (collection of rooted trees) is represented by the set of tree features, with each tree represented by concatenating its site-level features. With $z$ the rooted tree topology with root $u$, we represent the tree at site $i$ by its root level Fitch feature $f_u^i$ defined in §3.1.2. The proposition below, analogous to Proposition 1, shows the representation power of the proposed feature.

**Proposition 2.** *Let* $s_1 = \{z_1, z_2, \dots z_l\}$ *and* $s_2 = \{z_1', z_2', \dots z_l'\}$ *be two non-terminal states such that* $s_1 \neq s_2$ *but sharing the same Fitch features* $f_{z_i} = f_{z_i'} \; \forall i$. *Let* $\boldsymbol{a}$ *be any sequence of actions, which, applied to* $s_1$ *and* $s_2$, *respectively, results in tree topologies* $x, x' \in \mathcal{Z}$, *with two partial trajectories* $\tau = (s_1 \to \cdots \to x), \tau' = (s_2 \to \cdots \to x')$. *If* $P_F$ *is the policy of an optimal GFlowNet with uniform* $P_B$, *then* $P_F(\tau) = P_F(\tau')$

This shows that the Fitch features contain sufficient information for PhyloGFN-Parsimony. Furthermore, the Fitch features are used in the computation of the reward by Fitch's algorithm, so their use in the policy model does not introduce additional variables or extra computation.

**Temperature-conditioned PhyloGFN** The temperature $T$ controls the trade-off between sample diversity and parsimony scores. Following Zhang et al. (2023a), we extend PhyloGFN-Parsimony by conditioning the policy on $T$, with reward $R(z; T) = \exp\left(\frac{C - M(z|Y)}{T}\right)$, and we learn a sampler such that $P^\top(z; T) \propto R(z; T)$. See Appendix E for more details.

Table 1: Marginal log-likelihood estimation with different methods on real datasets DS1-DS8. PhyloGFN outperforms $\phi$-CSMC across all datasets and GeoPhy on most. *VBPI-GNN uses predefined tree topologies in training and is not directly comparable.

| | MCMC | | ML-based / amortized, full tree space | | |
|---|---|---|---|---|---|
| Dataset | MrBayes SS | VBPI-GNN* | $\phi$-CSMC | GeoPhy | PhyloGFN |
| DS1 | $-7108.42$ $_{\pm 0.18}$ | $-7108.41$ $_{\pm 0.14}$ | $-7290.36$ $_{\pm 7.23}$ | $-7111.55$ $_{\pm 0.07}$ | $\mathbf{-7108.95}$ $_{\pm 0.06}$ |
| DS2 | $-26367.57$ $_{\pm 0.48}$ | $-26367.73$ $_{\pm 0.07}$ | $-30568.49$ $_{\pm 31.34}$ | $\mathbf{-26368.44}$ $_{\pm 0.13}$ | $\mathbf{-26368.90}$ $_{\pm 0.28}$ |
| DS3 | $-33735.44$ $_{\pm 0.5}$ | $-33735.12$ $_{\pm 0.09}$ | $-33798.06$ $_{\pm 6.62}$ | $\mathbf{-33735.85}$ $_{\pm 0.12}$ | $\mathbf{-33735.6}$ $_{\pm 0.35}$ |
| DS4 | $-13330.06$ $_{\pm 0.54}$ | $-13329.94$ $_{\pm 0.19}$ | $-13582.24$ $_{\pm 35.08}$ | $-13337.42$ $_{\pm 1.32}$ | $\mathbf{-13331.83}$ $_{\pm 0.19}$ |
| DS5 | $-8214.51$ $_{\pm 0.28}$ | $-8214.64$ $_{\pm 0.38}$ | $-8367.51$ $_{\pm 8.87}$ | $-8233.89$ $_{\pm 6.63}$ | $\mathbf{-8215.15}$ $_{\pm 0.2}$ |
| DS6 | $-6724.07$ $_{\pm 0.86}$ | $-6724.37$ $_{\pm 0.4}$ | $-7013.83$ $_{\pm 16.99}$ | $-6733.91$ $_{\pm 0.57}$ | $\mathbf{-6730.68}$ $_{\pm 0.54}$ |
| DS7 | $-37332.76$ $_{\pm 2.42}$ | $-37332.04$ $_{\pm 0.12}$ | | $\mathbf{-37350.77}$ $_{\pm 11.74}$ | $-37359.96$ $_{\pm 1.14}$ |
| DS8 | $-8649.88$ $_{\pm 1.75}$ | $-8650.65$ $_{\pm 0.45}$ | $-9209.18$ $_{\pm 18.03}$ | $-8660.48$ $_{\pm 0.78}$ | $\mathbf{-8654.76}$ $_{\pm 0.19}$ |

### 4.3 MODEL ARCHITECTURE AND TRAINING

**Parameterization of forward transitions** We parameterize the forward transitions of tree topology construction using a Transformer-based neural network, whose architecture is shown in Fig. 1. We select Transformer because the input is a set and the model needs to be order-equivariant. For a state consisting of $n$ trees, after $n$ embeddings are generated from the Transformer encoder, $\binom{n}{2}$ pairwise features are created for all possible pairs of trees, and a common MLP generates probability logits for joining every tree pair. PhyloGFN-Bayesian additionally requires generating edge lengths. Once the pair of trees to join is selected, another MLP is applied to the corresponding pair feature to generate probability logits for sampling the edge lengths. See more details in D.

**Off-policy training** The action model $P_F(\cdot \mid s; \theta)$ is trained with the trajectory balance objective. Training data are generated from two sources: (i) A set of trajectories constructed from the currently trained GFlowNet, with actions sampled from the policy with probability $1 - \epsilon$ and uniformly at random with probability $\epsilon$. The $\epsilon$ rate drops from a pre-defined $\epsilon_{start}$ to near 0 during the course of training. (ii) Trajectories corresponding to the best trees seen to date (replay buffer). Trajectories are sampled backward from these high-reward trees with uniform backward policy.

**Temperature annealing** For PhyloGFN-Parsimony, it is crucial to choose the appropriate temperature $T$. Large $T$ defines a flat target distribution, while small $T$ makes the reward landscape less smooth and leads to training difficulties. We cannot predetermine the ideal choice of $T$ before training, as we do not know *a priori* the parsimony score for the dataset. Therefore, we initialize the training with large $T$ and reduce $T$ periodically during training. See Appendix E for details.

### 4.4 MARGINAL LOG-LIKELIHOOD ESTIMATION

To assess how well the GFlowNet sampler approximates the true posterior distribution, we use the following importance-weighted variational lower bound on the marginal log-likelihood (MLL):

$$\log P(Y) \geq \mathbb{E}_{\tau_1,\ldots,\tau_k \sim P_F} \log \left( P(z) \frac{1}{K} \sum_{\substack{\tau_i \\ \tau_i: s_0 \to \cdots \to (z_i, b_i)}}^{k} \frac{P_B(\tau_i|z_i, b_i) R(z_i, b_i)}{P_F(\tau_i)} \right). \tag{5}$$

Our estimator is computed by sampling a batch of $K$ trajectories and averaging $\frac{P(z) P_B(\tau|z,b) R(z,b)}{P_F(\tau)}$ over the batch. This expectation of this estimate is guaranteed to be a lower bound on $\log P(Y)$ and its bias decreases as $K$ grows (Burda et al., 2016).

PhyloGFN-Bayesian models branch lengths with discrete multinomial distributions, while in reality these are continuous variables. To properly estimate the MLL and compare with other methods defined over continuous space, we augment our model to a continuous sampler by performing random perturbations over edges of trees sampled from PhyloGFN-Bayesian. The perturbation follows the uniform distribution $\mathcal{U}_{[-0.5\omega, 0.5\omega]}$, where $\omega$ is the fixed bin size for edge modeling in PhyloGFN-Bayesian. The resulting model over branch lengths is then a piecewise-constant continuous distribution. We discuss the computation details as well as the derivation of (5) in Appendix B.

### 5 EXPERIMENTS

We evaluate PhyloGFN on a suite of 8 real-world benchmark datasets (Table S1 in Appendix C) that is standard in the literature. These datasets feature pre-aligned sequences and vary in difficulty (27

to 64 sequences; 378 to 2520 sites). In the following sections we present our results and analysis on Bayesian and parsimony-based phylogenetic inference.

## 5.1 BAYESIAN PHYLOGENETIC INFERENCE

PhyloGFN is compared with a variety of baselines in terms of sampling-based estimates of marginal log-likelihood (MLL; see details in §4.4). The baselines we compare to are MrBayes SS, Stepping-Stone sampling algorithm implemented in MrBayes (Ronquist et al., 2012), and three variational inference methods: VBPI-GNN (Zhang, 2023), $\phi$-CSMC proposed in VaiPhy (Koptagel et al., 2022), and Geo-Phy (Mimori & Hamada, 2023). The sampling setup for MrBayes follows Zhang & Matsen IV (2018b) and otherwise show the highest MLL reported in the respective papers. PhyloGFN MLL is estimated following the formulation in §4.4, with mean and standard deviation obtained from 10 repetitions, each using 1000 samples. See Sections E and G for additional training details, hardware specifications, and running time comparison. Additional results from two repeated experiments are in Table S5.

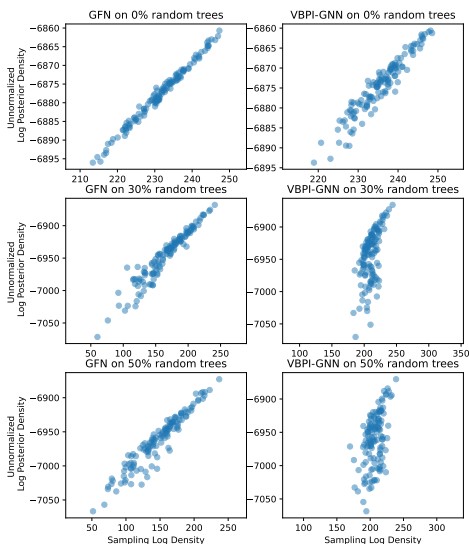

Figure 2: Model sampling log-density vs. unnormalized posterior log-density for high/medium/low-probability trees on DS1. We highlight that PhyloGFN-Bayesian performs significantly better on medium- and low-probability trees, highlighting its superiority in modeling the entire data space.

The results are summarized in Table S4. Our PhyloGFN is markedly superior to $\phi$-CSMC across all datasets and outperforms GeoPhy on most, with the exception of DS2 and DS3 where the two perform similarly, and DS7 where GeoPhy obtains a better result. VBPI-GNN, is the only machine learning-based method that performs on par against MrBayes, the current gold standard in Bayesian phylogenetic inference. However, it should be emphasized that VBPI-GNN requires a set of pre-defined tree topologies that are likely to achieve high likelihood, and as a consequence, its training and inference are both constrained to a small space of tree topologies. On the other hand, PhyloGFN operates on the full space of tree topologies and, in fact, achieves a closer fit to the true posterior distribution. To show this, for each dataset, we created three sets of phylogenetic trees with high/medium/low posterior probabilities and obtained the corresponding sampling probabilities from PhyloGFN and VBPI-GNN. The three classes of trees are generated from VBPI-GNN by randomly inserting uniformly sampled actions into its sequential tree topology construction process with 0%, 30%, or 50% probability, respectively, which circumvents VBPI-GNN's limitation of being confined to a small tree topology space. Table 2 and Fig. 2 show that PhyloGFN achieves higher Pearson correlation between the sampling log-probability and the unnormalized ground truth posterior log-density for the majority of datasets and classes of trees. In particular, while VBPI-GNN performs better on high-probability trees, its correlation drops significantly on lower-probability trees. On the other hand, PhyloGFN maintains a high correlation for all three classes of trees across all datasets, the only exception being the high-probability trees in DS7. See Appendix K for details and extended results and Appendix I for a short explanation of the significance of modeling suboptimal trees.

**Continuous branch length modeling** Following the initial submission, additional experiments involving continuous branch length modeling demonstrate PhyloGFN's ability to achieve state-of-the-art Bayesian inference performance. For more details, please see Appendix F.

## 5.2 PARSIMONY-BASED PHYLOGENETIC INFERENCE

As a special case of Bayesian phylogenetic inference, the parsimony problem is concerned with finding the most-parsimonious trees – a task which is also amenable to PhyloGFN. Here, we compare to the state-of-the-art parsimony analysis software PAUP* (Swofford, 1998). For all datasets, our PhyloGFN and PAUP* are able to identify the same set of optimal solutions to the Large Parsimony problem, ranging from a single optimal tree for DS1 to 21 optimal trees for DS8.

Although the results are similar between PhyloGFN and PAUP*, once again we emphasize that PhyloGFN is based on a rigorous mathematical framework of fitting and sampling from well-defined

Table 2: Pearson correlation of model sampling log-density and ground truth unnormalized posterior log-density for each dataset on high/medium/low posterior density trees generated by VBPI-GNN. PhyloGFN achieves a good fit on both high and low posterior density regions.

| Dataset | No random | | 30% random | | 50% random | |
|---|---|---|---|---|---|---|
| | PhyloGFN | VBPI-GNN | PhyloGFN | VBPI-GNN | PhyloGFN | VBPI-GNN |
| DS1 | **0.994** | 0.955 | **0.961** | 0.589 | **0.955** | 0.512 |
| DS2 | 0.930 | **0.952** | **0.948** | 0.580 | **0.919** | 0.538 |
| DS3 | 0.917 | **0.968** | **0.963** | 0.543 | **0.950** | 0.499 |
| DS4 | 0.942 | **0.960** | **0.945** | 0.770 | **0.966** | 0.76 |
| DS5 | **0.969** | 0.965 | **0.937** | 0.786 | **0.939** | 0.773 |
| DS6 | **0.993** | 0.887 | **0.973** | 0.816 | **0.934** | 0.702 |
| DS7 | 0.624 | **0.955** | **0.787** | 0.682 | **0.764** | 0.678 |
| DS8 | **0.978** | 0.955 | **0.913** | 0.604 | **0.901** | 0.463 |

posterior distributions over tree topologies, whereas PAUP* relies on heuristic algorithms. To put it more concretely, we show in Fig. S2 that PhyloGFN is able to (i) learn a smooth echelon of sampling probabilities that distinguish the optimal trees from suboptimal ones; (ii) learn similar sampling probabilities for trees within each group of equally-parsimonious trees; and (iii) fit all $2n - 3$ rooted trees that belong to the same unrooted tree equally well.

Finally, Fig. 3 shows that a single temperature-conditioned PhyloGFN can sample phylogenetic trees of varying ranges of parsimony scores by providing suitable temperatures $T$ as input to the model. Also, PhyloGFN is able to sample proportionately from the Boltzmann distribution defined at different temperatures and achieves high correlation between sampling log-probability and log-reward. Although the temperature-conditioned PhyloGFN has only been trained on a small range of temperatures between 4.0 and 1.0, Fig. 3 shows it can also approximate the distribution defined at temperature 8.0. Further results are presented in Appendix L.

## 6   DISCUSSION AND FUTURE WORK

In this paper, we propose PhyloGFN, a GFlowNet-based generative modeling algorithm, to solve parsimony-based and Bayesian phylogenetic inference. We design an intuitive yet effective tree construction procedure to efficiently model the entire tree topology space. We propose a novel tree representation based on Fitch's and Felsenstein's algorithms to represent rooted trees without introducing additional learnable parameters, and we show the sufficiency of our features for the purpose of phylogenetic tree generation. We apply our algorithm on eight real datasets, demonstrating that PhyloGFN is competitive with or superior to prior works in terms of marginal likelihood estimation, while achieving a closer fit to the target distribution compared to state-of-the-art variational inference methods. While our initial experiments with continuous branch length sampling have demonstrated notable performance enhancements, there remains a need for future research to address

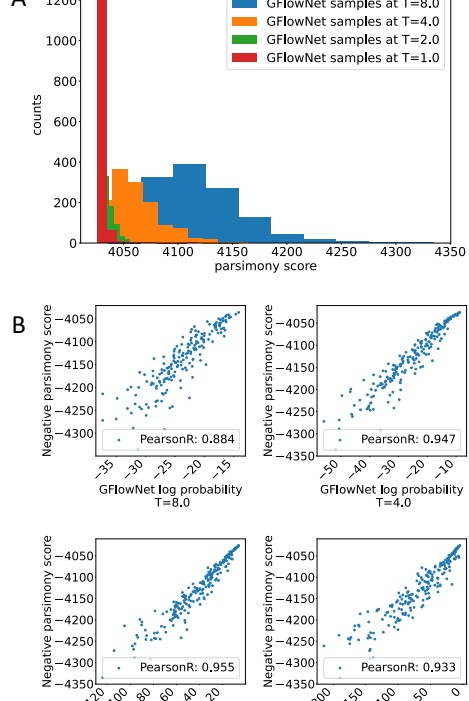

Figure 3: A temperature-conditioned PhyloGFN is trained on DS1 using temperatures sampled between 4.0 and 1.0. (A) Parsimony score distribution with varying statistical temperature input (8.0 to 1.0) to PhyloGFN policy (10,000 trees sampled per temperature). (B) PhyloGFN achieves high Pearson correlation at each temperature.

training efficiency. In addition, we plan to explore the use of conditional GFlowNets to amortize the dependence on the sequence dataset itself. This would allow a single trained GFlowNet to sample phylogenetic trees for sets of sequences that were not seen during training.

## REPRODUCIBILITY

Code and data are available `https://github.com/zmy1116/phylogfn`. See Appendix J for a detailed description.

## ACKNOWLEDGMENTS

We thank Oskar Kviman and Jens Lagergren for the insightful discussions on GFlowNet and phylogenetic inference. We thank Tianyu Xie and Cheng Zhang for discussions on continuous branch lengths modeling. We acknowledge the support of Mingyang Zhou by the NSERC Discovery grant awarded to Mathieu Blanchette, and the funding from CIFAR, NSERC, IBM, Intel, Genentech and Samsung to Yoshua Bengio. Finally, we thank the Digital Alliance of Canada for providing the computational resources.

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

## A  PROOFS OF PROPOSITIONS 1 AND 2

Before proving proposition 1, we first prove the following two lemmas. First, we show that for two states sharing the same tree features, applying the same action to the states results in two new states still sharing the same features.

**Lemma 1.** *Let $s_1 = \{(z_1, b_1), (z_2, b_2) \ldots (z_l, b_l)\}$ and $s_2 = \{(z'_1, b'_1), (z'_2, b'_2) \ldots (z'_l, b`_l)\}$ be two non-terminating states sharing the same features $\rho_i = \rho'_i$. Let a be the action that joins the trees with indices $(v, w)$ to form a new tree indexed u with edge lengths $(b(e_{uv}), b(e_{uw}))$. By applying a on $s_1$, we join $(z_v, b_v)$ and $(z_w, b_w)$ to form new tree $(z_u, b_u)$. By applying a on $s_2$, we join $(z'_v, b'_v)$ and $(z'_w, b'_w)$ to form new tree $(z'_u, b'_u)$. Then the new trees' features are equal: $\rho_u = \rho'_u$.*

*Proof.* We show that $\rho_u$ can be calculated from $\rho_v$ and $\rho_w$:

$$\rho_u^i[j] = P(b(e_{uv}))^{\frac{1}{m}} \times P(b(e_{uw}))^{\frac{1}{m}} \times \sum_{k \in \Sigma} P(a_u^i = j | a_v^i = k, b_z(e_{uv})) \rho_v^i[k]$$

$$\times \sum_k P(a_u^i = j | a_w^i = k, b(e_{uw})) \rho_w^i[k]$$

since $\rho_v = \rho'_v$, $\rho_w = \rho'_w$ and $(b(e_{uv}), b(e_{uw}))$ are new branch lengths for both two trees. Therefore $\rho_u = \rho'_u$ □

Next, we show that for two states sharing the same tree features, applying the same action sequences results in two phylogenetic trees with the same reward.

**Lemma 2.** *Let $s_1 = \{(z_1, b_1), (z_2, b_2) \ldots (z_l, b_l)\}$ and $s_2 = \{(z'_1, b'_1), (z'_2, b'_2) \ldots (z'_l, b'_l)\}$ be two non-terminating states sharing the same features $\rho_i = \rho'_i$. Let **a** be any sequence of actions to apply on $s_1$ and $s_2$ to form full trees $x = (z, b_z), x' = (z', b'_z), R(z, b) = R(z', b')$.*

*Proof.* Let $\rho_u$ denote the tree feature for $(z, b)$, $\rho_u^i = f_u^i \prod_e P(b(e))$. We first show that the reward can be directly calculated from the root feature $\rho_u$:

$$\prod_i P(a_{2n-1}) \cdot \rho_u = \prod_e P(b(e)) \prod_i P(a_{2n-1}) \cdot f_u^i$$
$$= P(b)P(Y|z, b)$$
$$= R(z, b),$$

where $P(a_{2n-1})$ is the constant root character assignment probability. As **a** is applied to $s_1$ and $s_2$ in a sequential manner, at every step we obtain two state swith the same tree features (by Lemma 1), until, at the final state, $\rho_u = \rho'_u$. It follows that $R(z, b) = R(z', b')$. □

We are now ready to prove the propositions.

**Proposition 1.** *Let $s_1 = \{(z_1, b_1), (z_2, b_2) \ldots (z_l, b_l)\}$ and $s_2 = \{(z'_1, b'_1), (z'_2, b'_2) \ldots (z'_l, b'_l)\}$ be two non-terminal states such that $s_1 \neq s_2$ but sharing the same features $\rho_i = \rho'_i$. Let **a** be any sequence of actions, which applied to $s_1$ and $s_2$, respectively, results in full weighted trees $x = (z, b_z), x' = (z', b')$, with two partial trajectories $\tau = (s_1 \to \cdots \to x), \tau' = (s_2 \to \cdots \to x')$. If $P_F$ is the policy of an optimal GFlowNet with uniform $P_B$, then $P_F(\tau) = P_F(\tau')$.*

*Proof of Proposition 1.* Let $\mathcal{G}_{s_1}, \mathcal{G}_{s_2}$ be sub-graphs of the GFlowNet state graph $\mathcal{G} = (\mathcal{S}, \mathcal{A})$ defined by reachable states from $s_1$ and $s_2$ in $\mathcal{G}$. Since $s_1$ and $s_2$ have the same number of trees, $\mathcal{G}_{s_1}$ and $\mathcal{G}_{s_2}$ have the graph structure. Let $X_1, X_2 \subseteq X$ be the terminal states reachable from $s_1$ and $s_2$. There is thus a bijective correspondence between $X_1$ and $X_2$: for every action set **a** applying on $s_1$ to obtain $x \in X_1$, we obtain $x' \in X_2$ by applying **a** on $s_2$. Let $\tau$ and $\tau'$ be the partial trajectories created by applying **a** on $s_1$ and $s_2$, $P_B(\tau|x) = P_B(\tau'|x')$. Moreover, $R(x) = R(x')$ since $s_1$ and $s_2$ share the same set of features. We have the following expressions for $P_F(\tau)$ and $P_F(\tau')$:

$$P_F(\tau) = \frac{R(x)P_B(\tau|x)}{\sum_{\tau_j, x_j} R(x_j)P_B(\tau_j|x_j)}, \quad P_F(\tau') = \frac{R(x')P_B(\tau'|x')}{\sum_{\tau'_j, x'_j} R(x'_j)P_B(\tau'_j|x'_j)}.$$

Hence $P_F(\tau) = P_F(\tau')$.

□

**Proposition 2.** *Let $s_1 = \{z_1, z_2, \ldots z_l\}$ and $s_2 = \{z'_1, z'_2, \ldots z'_l\}$ be two non-terminal states such that $s_1 \neq s_2$ but sharing the same Fitch features $\boldsymbol{f}_{z_i} = \boldsymbol{f}_{z'_i} \forall i$. Let $\boldsymbol{a}$ be any sequence of actions, which, applied to $s_1$ and $s_2$, respectively, results in tree topologies $x, x' \in \mathcal{Z}$, with two partial trajectories $\tau = (s_1 \rightarrow \cdots \rightarrow x), \tau' = (s_2 \rightarrow \cdots \rightarrow x')$. If $P_F$ is the policy of an optimal GFlowNet with uniform $P_B$, then $P_F(\tau) = P_F(\tau')$*

*Proof of Proposition 2.* We use the same notation as in the proof of of Proposition 1. Since $s_1$ and $s_2$ have the same number of trees, $\mathcal{G}_{s_1}$ and $\mathcal{G}_{s_2}$ have the same graph structure. Let $\mathcal{X}_1, \mathcal{X}_2 \subseteq \mathcal{X}$ be the terminal states reachable from $s_1$ and $s_2$. There is a bijective correspondence between $\mathcal{X}_1$ and $\mathcal{X}_2$: for every action set $\boldsymbol{a}$ applying on $s_1$ to obtain $x \in \mathcal{X}_1$, we obtain $x' \in \mathcal{X}_2$ by applying $\boldsymbol{a}$ on $s_2$. Let $\tau$ and $\tau'$ be the partial trajectories created by applying $\boldsymbol{a}$ on $s_1$ and $s_2$, $P_B(\tau|x) = P_B(\tau'|x')$.

For simplicity, we denote $M(x) = M(x|L(x))$. It is worth to note that for two tree topologies sharing the same Fitch feature, their parsimony scores do not necessarily equal. However, when applying $\boldsymbol{a}$ on two states $s_1$ and $s_2$ sharing the Fitch feature, the additional parsimony scores introduced are the same:

$$M(x) - \sum_i M(z_i) = M(x') - \sum_i M(z'_i)$$

We have the following expressions for $P_F(\tau)$ and $P_F(\tau')$:

$$P_F(\tau) = \frac{e^{\frac{-M(x)}{T}} P_B(\tau|x)}{\sum_{\tau_j, x_j} e^{\frac{-M(x_j)}{T}} P_B(\tau_j|x_j)}, \quad P_F(\tau') = \frac{e^{\frac{-M(x')}{T}} P_B(\tau|x')}{\sum_{\tau'_j, x'_j} e^{\frac{-M(x'_j)}{T}} P_B(\tau'_j|x'_j)}$$

We multiply $\frac{e^{\frac{\sum_i M(z_i)}{T}}}{e^{\frac{\sum_i M(z_i)}{T}}}$ by $P_F(\tau)$ and $\frac{e^{\frac{\sum_i M(z'_i)}{T}}}{e^{\frac{\sum_i M(z'_i)}{T}}}$ by $P_F(\tau')$ and obtain:

$$P_F(\tau) = \frac{e^{\frac{\sum_i M(z_i) - M(x)}{T}} P_B(\tau|x)}{\sum_{\tau_j, x_j} e^{\frac{\sum_i M(z_i) - M(x_j)}{T}} P_B(\tau_j|x_j)}, \quad P_F(\tau') = \frac{e^{\frac{\sum_i M(z'_i) - M(x')}{T}} P_B(\tau|x')}{\sum_{\tau'_j, x'_j} e^{\frac{\sum_i M(z'_i) - M(x'_j)}{T}} P_B(\tau'_j|x'_j)}$$

Since $e^{\frac{\sum_i M(z'_i) - M(x'_j)}{T}} P_B(\tau'_j|x'_j) = e^{\frac{\sum_i M(z_i) - M(x_j)}{T}} P_B(\tau_j|x_j)$ for all $j$, $P_F(\tau) = P_F(\tau')$. $\square$

## B  MARGINAL LOG-LIKELIHOOD ESTIMATION

**Estimating the sampling likelihood of a terminal state**  In the PhyloGFN state space (in both the Bayesian and parsimony-based settings), there exist multiple trajectories leading to the same terminal state $x$, hence the sampling probability of $x$ is calculated as: $P^\top(x) = \sum_{\tau:s_0 \rightarrow \cdots \rightarrow x} P_F(\tau)$. This sum is intractable for large-scale problems. However, we can estimate the sum using importance sampling(Zhang et al., 2022):

$$P_F^\top(x) \approx \frac{1}{K} \sum_{\tau_i:s_0 \rightarrow \cdots \rightarrow x} \frac{P_F(\tau_i)}{P_B(\tau_i|x)}, \tag{S1}$$

where the trajectories $\tau_i$ are sampled from $P_B(\tau_i \mid x)$. The logarithm of the right side of (S1) is, in expectation, a lower bound on the logarithm of the true sampling likelihood on the left side of (S1).

**Estimating the MLL**  The lower bound on MLL can be evaluated using the importance-weighted bound $\log P(\boldsymbol{Y}) \geq \mathbb{E}_{x_1 \ldots x_k \sim P_F} \log \frac{1}{K} \sum \frac{P(x, \boldsymbol{Y})}{P^\top(x)}$(Burda et al., 2016). However, we cannot use it for PhyloGFN since we cannot compute the exact $P^T(x)$, only get a lower bound on it using (S1). Therefore, we propose the following variational lower bound:

$$\log P(\boldsymbol{Y}) \geq \mathbb{E}_{\tau_1, \ldots, \tau_k \sim P_F} \log \left( P(z) \frac{1}{K} \sum_{\substack{\tau_i \\ \tau_i:s_0 \rightarrow \cdots \rightarrow (z_i, b_i)}}^{k} \frac{P_B(\tau_i|z_i, b_i) R(z_i, b_i)}{P_F(\tau_i)} \right)$$

We show the derivation of the estimator and thus prove its consistency:

$$
\begin{aligned}
P(Y) &= \int_{z,b} P(Y|z,b)P(b|z)P(z) \\
&= \int_{z,b} R(z,b)P(z) \\
&= \int_{z,b} R(z,b)P(z) \sum_{\tau:s_0\ldots x=(z,b)} P_B(\tau|z,b) \\
&= \int_{z,b} R(z,b)P(z) \sum_{\tau:s_0\ldots x=(z,b)} P_B(\tau|z,b)\frac{P_F(\tau)}{P_F(\tau)} \\
&= \int_{z,b} \sum_{\tau:s_0\ldots x=(z,b)} P_F(\tau)\frac{P_B(\tau|z,b)R(z,b)P(z)}{P_F(\tau)} \\
&= \int_{\tau:s_0\ldots x_\tau=(z_\tau,b_\tau)} P_F(\tau)\frac{P_B(\tau|z_\tau,b_\tau)R(z_\tau,b_\tau)P(z)}{P_F(\tau)} \\
&= P(z)\mathbb{E}_{\tau\sim P_F}\frac{P_B(\tau|z_\tau,b_\tau)R(z_\tau,b_\tau)}{P_F(\tau)} \\
&\approx P(z)\frac{1}{K}\sum_{\substack{\tau_i\sim P_F \\ \tau_i:s_0\ldots(z_i,b_i)}}^{k}\frac{P_B(\tau_i|z_i,b_i)R(z_i,b_i)}{P_F(\tau_i)}.
\end{aligned}
$$

One can show, in a manner identical to the standard importance-weighted bound, that this estimate is a lower bound on $\log P(Y)$.

PhyloGFN-Bayesian models edge lengths using discrete distributions. To estimate our algorithm's MLL, we augment the sampler to a continuous sampler by modeling branch lengths with a piecewise-constant continuous distribution based on the fixed-bin multinomial distribution of PhyloGFN. We can still use the above formulation to estimate the lower bound. However, each trajectory now has one extra step: $\tau' = s_0 \to \cdots \to (z,b) \to (z,b_i)$ where $(z,b_i)$ is obtained from $z,b$ by randomly perturbing each branch length by adding noise from $U[-0.5\omega, 0.5\omega]$, where $\omega$ is the bin size used for PhyloGFN. Let $\tau = s_0 \to \cdots \to (z,b)$ be the original trajectory in the discrete PhyloGFN, we can compute $P_F(\tau'), P_B(\tau')$ from $P_F(\tau), P_B(\tau)$:

$$
P_F(\tau') = P_F(\tau)\frac{1}{\omega^{|E(z)|}}, \quad P_B(\tau') = P_B(\tau)
$$

The term $\frac{1}{\omega^{|E(z)|}}$ is introduced in $P_F$ because we additionally sample over a uniform range $\omega$ for all $|E(z)|$ edges. The backward probability $P_B$ stays unchanged because given $(z,b)$ generated from the discrete GFN, for any $(z,b')$ resulting from perturbing edges, $(z,b)$ is the the only possible ancestral state.

## C   Dataset information

Table S1: Statistics of the benchmark datasets from DS1 to DS8.

| Dataset | # Species | # Sites | Reference |
|---------|-----------|---------|-----------|
| DS1 | 27 | 1949 | Hedges et al. (1990) |
| DS2 | 29 | 2520 | Garey et al. (1996) |
| DS3 | 36 | 1812 | Yang & Yoder (2003) |
| DS4 | 41 | 1137 | Henk et al. (2003) |
| DS5 | 50 | 378 | Lakner et al. (2008) |
| DS6 | 50 | 1133 | Zhang & Blackwell (2001) |
| DS7 | 59 | 1824 | Yoder & Yang (2004) |
| DS8 | 64 | 1008 | Rossman et al. (2001) |

## D   Modeling

Given the character set $\Sigma$, we use one-hot encoding to represent each site in a sequence. To deal with wild characters in the dataset, for parsimony analysis we consider them as one special character in $\Sigma$, while in Bayesian inference, we represent them by a vector of $\mathbb{1}$.

For both PhyloGFN-Parsimony and PhyloGFN-Bayesian, given a state with $l$ rooted trees, its representation feature is the set $\{\rho_1 \ldots \rho_l\}$, where $\rho$ is a vector of length $m|\Sigma|$. For example, for DNA sequences of 1000 sites, each $\rho$ would have length 4000. Therefore, before passing these features to the Transformer encoder, we first use a linear transformation to obtain lower-dimensional embeddings of the input features.

We use the Transformer architecture (Vaswani et al., 2017) to build the **Transformer encoder**. For a state with $l$ trees, the output is a set of $l+1$ features $\{e_s, e_1, \ldots, e_l\}$ where $e_s$ denotes the summary feature (*i.e.*, the [CLS] token of the Transformer encoder input).

To select pairs of trees to join, we evaluate tree-pair features for every pair of trees in the state and pass these tree-pair features as input to the **tree MLP** to generate probability logits for all pairs of trees. The tree-pair feature for a tree pair $(i, j)$ with representations $e_i$, $s_j$ is the concatenation of $e_i + e_j$ with the summary embedding of the state, *i.e.*, the feature is $[e_s; e_i + e_j]$, where $[\cdot; \cdot]$ denotes vector direct sum (concatenation). For a state with $l$ trees, $\binom{l}{2} = \frac{l(l-1)}{2}$ such pairwise features are generated for all possible pairs.

To generate edge lengths for the joined tree pair $(i, j)$, we pass $[e_s; e_i; e_j]$ – the concatenation of the summary feature with the tree-level features of trees $i$ and $j$ – as input to the **edge MLP**. For unrooted tree topologies we need to distinguish two scenarios: (i) when only two rooted trees are left in the state (*i.e.*, at the last step of PhyloGFN state transitions), we only need to generate a single edge; and (ii) when there are more than two rooted trees in the state, a pair of edges is required. Therefore, two separate edge MLPs are employed, as edge length is modeled by $k$ discrete bins, the edge MLP used at the last step has an output size of $k$ (to model a distribution over a single edge length) whereas the other edge MLP would have an output size of $k^2$ (to model a joint distribution over two edge lengths).

For the temperature-conditioned PhyloGFN, as then temperature $T$ is passed to our PhyloGFN as an input, two major modifications are required: (i) the estimation of the partition $Z_\theta$ is now a function of $T$: $Z_\theta(T)$, which is modeled by a **Z MLP**; (ii) the summary token to the Transformer encoder also captures the temperature information by replacing the usual [CLS] token with a **temp MLP** that accepts $T$ as input.

## E   Training details

Here, we describe the training details for our PhyloGFN. For PhyloGFN-Bayesian, our models are trained with fixed 500 epochs. For PhyloGFN-Parsimony, our models are trained until the probability of sampling the optimal trees, or the most parsimonious trees our PhyloGFN has seen so far, is above 0.001. Each training epoch consists of 1000 gradient update steps using a batch of 64 trajectories. For $\epsilon$-greedy exploration, the $\epsilon$ value is linearly annealed from 0.5 to 0.001 during the first 240 epochs. All common hyperparameters for PhyloGFN-Bayesian and PhyloGFN-Parsimony are shown in Table S3.

**Temperature annealing** For PhyloGFN-Bayesian, the initial temperature is set to 16 for all experiments. For PhyloGFN-Parsimony, $T$ is initialized at 4. Under the cascading temperature annealing scheme, $T$ is reduced by half per every 80 epochs of training. For PhyloGFN-Bayesian, $T$ is always reduced to and then fixed at 1, whereas for PhyloGFN-Parsimony, $T$ is only reduced when the most parsimonious trees seen by our PhyloGFN so far are sampled with a probability below 0.001.

**Hyperparameter $C$ selection** For PhyloGFN-Parsimony, the reward is defined as $R(x) = \exp\left(\frac{C-M(x|Y)}{T}\right)$, where $C$ is a hyperparameter introduced for training stability and it controls the magnitude of the partition function $Z = \sum_x R(x)$. Given that we cannot determine the precise value of $C$ prior to training since we do know the value of $Z$ as a priori, we use the following heuristic to choose $C$: 1000 random tree topologies are generated via stepwise-addition, and we set $C$ to the 10% quantile of the lowest parsimony score.

Similarly, $C$ is employed for PhyloGFN-Bayesian under the same goal of stabilizing the training and reducing the magnitude of the partition function. Recall the reward function $R(z, b)$ defined in §4.1, it can be rewritten as $R(z, b) = \exp\left(\frac{C-(-\log P(Y|z,b)P(b))}{T}\right)$ when $T = 1$. Note that $\exp\left(\frac{C}{T}\right)$ can be absorbed into the partition function. For selecting the $C$, once again we randomly sample 1000 weighted phylogenetic trees via stepwise-addition and with random edge length, followed by setting $C$ to the 10% quantile of the lowest $-\log P(Y|z, b)P(b)$.

**Temperature-conditioned PhyloGFN-Parsimony** The temperature-conditioned PhyloGFN is introduced so that a single trained PhyloGFN can be used to sample from a series of reward distributions defined by different $T$. We modify the fixed-temperature PhyloGFN-Parsimony by introducing $T$ as input in 3 places: (i) the reward function $R(x; T)$; (ii) the forward transition policy $P_F(x; T)$; and (iii) the learned partition function estimate $Z_\theta(T)$. To train the temperature-conditioned PhyloGFN, the TB objective also needs to be updated accordingly:

$$\mathcal{L}_{\text{TB}}(\tau; T) = \left(\log \frac{Z_\theta(T) \prod_{i=0}^{n-1} P_F(s_{i+1} \mid s_i; \theta, T)}{R(x, T)P_B(\tau \mid x)}\right)^2, \quad P_B(\tau \mid x) := \prod_{i=1}^n \frac{1}{|\text{Pa}(s_i)|},$$

Note that $P_B$ is unaffected.

During training, values for $T$ are randomly selected from the range $[T_{\min}, T_{\max}]$. When training with a state from the replay buffer, the temperature used for training is resampled and may be different than the one originally used when the state was added to the buffer.

We also employ a scheme of gradually reducing the average of sampled $T$ values through the course of training: $T$'s are sampled from a truncated normal distribution with a fixed pre-defined variance and a moving mean $T_\mu$ that linearly reduces from $T_{\max}$ to $T_{\min}$.

**Modeling branch lengths with discrete multinomial distribution** When a pair of trees are joined at the root, the branch lengths of the two newly formed edges are modeled jointly by a discrete multinomial distribution. The reason for using a joint distribution is because under the Jukes-Cantor evolution model, the likelihood at the root depends on the sum of the two branch lengths.

We use a different set of maximum edge length, bin number and bin size depending on each dataset, and by testing various combinations we have selected the set with optimal performance. The maximum branch length is chosen among $\{0.05, 0.1, 0.2\}$, and bin size $\omega$ is chosen among $\{0.001, 0.002, 0.004\}$. Table S2 shows the our final selected bin size and bin number for each dataset.

Table S2: Bin sizes and bin numbers used to model branch lengths for DS1 to DS8.

| Dataset | Bin Size | # Bins |
|---------|----------|--------|
| DS1 | 0.001 | 50 |
| DS2 | 0.004 | 50 |
| DS3 | 0.004 | 50 |
| DS4 | 0.002 | 100 |
| DS5 | 0.002 | 100 |
| DS6 | 0.001 | 100 |
| DS7 | 0.001 | 200 |
| DS8 | 0.001 | 100 |

Table S3: Common hyperparameters for PhyloGFN-Bayesian and PhyloGFN-Parsimony.

| Transformer encoder | |
| --- | --- |
| hidden size | 128 |
| number of layers | 6 |
| number of heads | 4 |
| learning rate (model) | 5e-5 |
| learning rate (Z) | 5e-3 |
| tree MLP | |
| hidden size | 256 |
| number of layers | 3 |
| edge MLP | |
| hidden size | 256 |
| number of layers | 3 |
| Z MLP (in temperature-conditioned PhyloGFN) | |
| hidden size | 128 |
| number of layers | 1 |
| temp MLP (in temperature-conditioned PhyloGFN) | |
| hidden size | 256 |
| number of layers | 3 |

# F  CONTINUOUS BRANCH LENGTH MODELING WITH PHYLOGFN

While the original PhyloGFN-Bayesian only samples discrete branch lengths, we have experimented with a new version of PhyloGFN that uses mixture of Gaussian to model and sample branch lengths — effectively treating them as continuous variables. We refer to this new version as PhyloGFN-Continuous and we point out that the edge modeling is the only implementational detail that differs from the previous models.

Specifically, the **edge MLP** of PhyloGFN-Continuous now outputs the parameters of the Gaussian mixture which models the logarithm of branch length. These include (1) logits for selecting the components of the Gaussian mixture, (2) mean of the log branch length in each mixture, and (3) log variance of the log branch length in each mixture.

Although PhyloGFN-Continuous continues to employ two separate **edge MLP**, one for the last step of state transition where only a single edge needs to be sampled and the other for sampling a pair of edges at the intermediate step, PhyloGFN-Continuous now samples each edge independently as we have found this to benefit training stability.

It is also worth pointing out that we no longer perform $\epsilon$-greedy random exploration on branch lengths, again for the sake of training stability, and the mean of the log branch length is initialized at $-4.0$ instead of $0.0$, which is a more reasonable value for log branch length. There are five components in each Gaussian mixture.

Results are shown below. We are delighted to report that PhyloGFN-Continuous has eliminated the quantization error introduced in the earlier discrete PhyloGFN-Bayesian and therefore, our new model is effectively performing on par with the state of the art MrBayes and VBPI-GNN models. In particular, PhyloGFN-Continuous has the smallest standard deviation for the MLL estimation across all datasets, with the only exception being DS7 where PhyloGFN-Continuous closely matches the performance of VBPI-GNN.

Note that PhyloGFN-Continuous has undergone the same training routine that is described in §E, as is PhyloGFN-Bayesian.

Table S4: Marginal log-likelihood estimation with different methods on real datasets DS1-DS8. PhyloGFN-C(ontinuous) now outperforms $\phi$-CSMC, GeoPhy and PhyloGFN-B(ayesian) across all datasets and it is effectively performing on par with the state of the arts MrBayes and VBPI-GNN.

| Dataset | MCMC | | ML-based / amortized, full tree space | | | |
| | MrBayes SS | VBPI-GNN* | $\phi$-CSMC | GeoPhy | PhyloGFN-B | PhyloGFN-C |
|---|---|---|---|---|---|---|
| DS1 | $-7108.42_{\pm 0.18}$ | $-7108.41_{\pm 0.14}$ | $-7290.36_{\pm 7.23}$ | $-7111.55_{\pm 0.07}$ | $-7108.95_{\pm 0.06}$ | $-7108.40_{\pm 0.04}$ |
| DS2 | $-26367.57_{\pm 0.48}$ | $-26367.73_{\pm 0.07}$ | $-30568.49_{\pm 31.34}$ | $-26368.44_{\pm 0.13}$ | $-26368.90_{\pm 0.28}$ | $-26367.70_{\pm 0.04}$ |
| DS3 | $-33735.44_{\pm 0.5}$ | $-33735.12_{\pm 0.09}$ | $-33798.06_{\pm 6.62}$ | $-33735.85_{\pm 0.12}$ | $-33735.6_{\pm 0.35}$ | $-33735.11_{\pm 0.02}$ |
| DS4 | $-13330.06_{\pm 0.54}$ | $-13329.94_{\pm 0.19}$ | $-13582.24_{\pm 35.08}$ | $-13337.42_{\pm 1.32}$ | $-13331.83_{\pm 0.19}$ | $-13329.91_{\pm 0.02}$ |
| DS5 | $-8214.51_{\pm 0.28}$ | $-8214.64_{\pm 0.38}$ | $-8367.51_{\pm 8.87}$ | $-8233.89_{\pm 6.63}$ | $-8215.15_{\pm 0.2}$ | $-8214.38_{\pm 0.16}$ |
| DS6 | $-6724.07_{\pm 0.86}$ | $-6724.37_{\pm 0.4}$ | $-7013.83_{\pm 16.99}$ | $-6733.91_{\pm 0.57}$ | $-6730.68_{\pm 0.54}$ | $-6724.17_{\pm 0.10}$ |
| DS7 | $-37332.76_{\pm 2.42}$ | $-37332.04_{\pm 0.12}$ | | $-37350.77_{\pm 11.74}$ | $-37359.96_{\pm 1.14}$ | $-37331.89_{\pm 0.14}$ |
| DS8 | $-8649.88_{\pm 1.75}$ | $-8650.65_{\pm 0.45}$ | $-9209.18_{\pm 18.03}$ | $-8660.48_{\pm 0.78}$ | $-8654.76_{\pm 0.19}$ | $-8650.46_{\pm 0.05}$ |

## G    RUNNING TIME AND HARDWARE REQUIREMENTS

PhyloGFN is trained on virtual machines equipped with 10 CPU cores and 10GB RAM for all datasets. We use one V100 GPU for datasets DS1-DS6 and one A100 GPU for DS7-DS8, although the choice of hardware is not essential for running our training algorithms.

For Bayesian inference, the models used for the MLL estimation in Table S4 are trained on a total of 32 million examples, with a training wall time ranging from 3 to 8 days across the eight datasets. However, our algorithm demonstrates the capacity to achieve similar performance levels with significantly reduced training data. The table S5 below presents the performance of PhyloGFN with 32 million training examples (**PhyloGFN Full**) and with only 40% of the training trajectories (**PhyloGFN Short**). Each type of experiment is repeated 3 times. Table S6 compares running time of the full experiment with the shorter experiment. The tables show that the shorter runs exhibit comparable performance to our full run experiments, and all conclude within 3 days.

We compare the running time of PhyloGFN with VI baselines (VBPI-GNN, Vaiphy, and GeoPhy) using the DS1 dataset. VBPI-GNN and GeoPhy are trained using the same virtual machine configuration as PhyloGFN (10 cores, 10GB ram, 1xV100 GPU). The training setup for both algorithms mirrors the one that yielded the best performance as documented in their respective papers. As for VaiPhy, we employed the recorded running time from the paper on a machine with 16 CPU cores and 96GB RAM. For PhyloGFN, we calculate the running time of the full training process (PhyloGFN-Full) and four shorter experiments with 40%, 24%, 16% and 8% training examples. The table S7 documents both the running time and MLL estimation for each experiment. While our most comprehensive experiment, PhyloGFN Full, takes the longest time to train, our shorter runs – all of which conclude training within a day – show only a marginal degradation in performance. Remarkably, even our shortest run, PhyloGFN - 8%, outperforms both GeoPhy and $\phi$-CSMC, achieving this superior performance with approximately half the training time of GeoPhy.

Table S5: PhyloGFN-Bayesian MLL estimation on 8 datasets. We repeat both full experiment and short experiment (with 40% training examples) 3 times

| Experiment | PhyloGFN Full | | | PhyloGFN Short (40% Training data) | | |
|---|---|---|---|---|---|---|
| Repeat | 1 | 2 | 3 | 1 | 2 | 3 |
| DS1 | -7108.95 ±0.06 | -7108.97 ±0.05 | -7108.94 ±0.05 | -7108.97 ±0.14 | -7108.94 ±0.22 | -7109.04 ±0.08 |
| DS2 | -26368.9 ±0.28 | -26368.77 ±0.43 | -26368.89 ±0.29 | -26368.9 ±0.39 | -26369.03 ±0.31 | -26368.88 ±0.32 |
| DS3 | -33735.6 ±0.35 | -33735.60 ±0.40 | -33735.68 ±0.64 | -33735.9 ±0.91 | -33735.83 ±0.62 | -33735.76 ±0.75 |
| DS4 | -13331.83 ±0.19 | -13331.80 ±0.31 | -13331.94 ±0.42 | -13332.04 ±0.57 | -13331.87 ±0.31 | -13331.78 ±0.37 |
| DS5 | -8215.15 ±0.2 | -8214.92 ±0.27 | 8214.85 ±0.28 | -8215.38 ±0.27 | -8215.37 ±0.26 | -8215.38 ±0.25 |
| DS6 | -6730.68 ±0.54 | -6730.72 ±0.26 | -6730.89 ±0.22 | -6731.35 ±0.31 | -6731.2 ±0.4 | -6731.1 ±0.38 |
| DS7 | -37359.96 ±1.14 | -37360.59 ±1.62 | -37361.51 ±2.89 | -37362.03 ±5.2 | -37363.43 ±2.2 | -37362.37 ±2.65 |
| DS8 | -8654.76 ±0.19 | -8654.67 ±0.39 | -8654.86 ±0.15 | -8655.8 ±0.95 | -8655.65 ±0.37 | -8654.96 ±0.46 |

Table S6: PhyloGFN-Bayesian training time

| Dataset | PhyloGFN Full | PhyloGFN Short |
|---|---|---|
| DS1 | 62h40min | 20h40min |
| DS2 | 69h16min | 28h |
| DS3 | 80h20min | 35h40min |
| DS4 | 103h54min | 44h30min |
| DS5 | 127h50min | 51h40min |
| DS6 | 135h10min | 53h10min |
| DS7 | 174h3min | 60h20min |
| DS8 | 190h25min | 61h40min |

## H    ABLATION STUDY

### H.1    BRANCH LENGTHS MODEL HYPERPARAMETERS

PhyloGFN models branch lengths using discrete multinomial distributions. When estimating MLL, the branch length model is transformed into a piecewise-constant continuous form, introducing a small quantization error. Two hyperparameters define the multinomial distribution: bin size and bin number. This analysis investigates the impact of bin size and bin number on MLL estimation.

Table S7: Running time of PhyloGFN-Bayesian and VI baseline methods on DS1

|                | Running Time | MLL |
|----------------|--------------|-----|
| VBPI-GNN       | 16h10min     | -7108.41 $_{\pm 0.14}$ |
| GeoPhy         | 12h50min     | -7111.55 $_{\pm 0.07}$ |
| $\phi$-CSMC**  | ~2h          | -7290.36 $_{\pm 7.23}$ |
| PhyloGFN - Full | 62h40min    | -7108.97 $_{\pm 0.05}$ |
| PhyloGFN - 40% | 20h40min     | -7108.97 $_{\pm 0.14}$ |
| PhyloGFN - 24% | 15h40min     | -7109.01 $_{\pm 0.15}$ |
| PhyloGFN - 16% | 10h50min     | -7109.15 $_{\pm 0.23}$ |
| PhyloGFN - 8%  | 5h10min      | -7110.65 $_{\pm 0.39}$ |

For the fixed bin size of 0.001, we assess four sets of bin numbers: 50, 100, 150, and 200, and for the fixed bin number of 50, we evaluate three sets of bin sizes: 0.001, 0.002, and 0.004. For each setup, we train a PhyloGFN-Bayesian model on the dataset DS1 using 12.8 millions training examples.

Table S8 displays the MLL obtained in each setup. A noticeable decline in MLL estimation occurs as the bin size increases, which is expected due to the increased quantization error. However, MLL estimation also significantly declines as the number of bins increases over 100 under the fixed bin size of 0.001. We conjecture this is because the size of the action pool for sampling the pair of branch lengths increases quadratically by the number of bins. For example at 200 bins, the MLP head that generates branch lengths has 40,000 possible outcomes, leading to increased optimization challenges.

### H.2 EXPLORATION POLICIES

PhyloGFN employs the following techniques to encourage exploration during training:

- $\epsilon$-greedy annealing: a set of trajectories are generated from the GFlowNet that is being trained, with actions sampled from the GFlowNet policy with probability $1 - \epsilon$ and uniformly at random with probability $\epsilon$. The $\epsilon$ rate drops from a pre-defined $\epsilon_{start}$ to near 0 through the course of training.

- Replay Buffer: a replay buffer is used to store the best trees seen to date, and to use them for training, random trajectories are constructed from these high-reward trees using the backward probability $P_B$.

- Temperature annealing: the training of GFlowNet begins at a large temperature $T$ and it is divided in half periodically during training.

To assess the effectiveness of various exploration methods, we train PhyloGFN-Bayesian on DS1 with the following setups:

1. All trajectories are generated strictly on-policy training without any exploration methods (On-policy).

2. All trajectories are generated with epsilon-greedy annealing ($\epsilon$).

3. Half of trajectories are generated with $\epsilon$-greedy annealing, and the other half are constructed from the replay buffer ($\epsilon$ + RP).

4. The same setup as 3, with the addition of temperature annealing. $T$ is initialized at 16 and reduced by half per every 1.5 million training examples until reaching 1 ($\epsilon$ + RP + T Cascading).

5. The same setup as 4, except the temperature drops linearly from 16 to 1 ($\epsilon$ + RP + T Linear).

For each setup, we train a model using 12.8 millions training examples. Table S9 displays the MLL estimation for each setup. On-policy training without any additional exploration strategies results in the poorest model performance. Epsilon-greedy annealing significantly enhances performance, and combining all three strategies yields the optimal results. There is no significant difference between cascading and linear temperature drops.

### I SIGNIFICANCE OF MODELING SUBOPTIMAL TREES WELL

Several downstream tasks that rely on phylogenetic inference require not only the identification of optimal (maximum likelihood or most parsimonious) trees, but also the proper sampling of suitably

Table S8: MLL estimation of PhyloGFN-Bayesian on DS1 with different combinations of bin size and bin number to model edge lengths.

| Bin Size | Bin Number | MLL |
|---|---|---|
| 0.001 | 50 | -7108.98 ±0.14 |
| 0.001 | 100 | -7108.96 ±0.18 |
| 0.001 | 150 | -7109.17 ±1.14 |
| 0.001 | 200 | -7109.3 ±7.23 |
| 0.002 | 50 | -7109.95 ±0.29 |
| 0.004 | 50 | -7114.48 ±0.52 |

Table S9: MLL estimation of PhyloGFN-Bayesian on DS1 with different exploration policies

| Training Policies | MLL |
|---|---|
| On policy | -7307.99 ±0.26 |
| $\epsilon$ | -7109.92 ±0.18 |
| $\epsilon$ + RP | -7109.95 ±0.22 |
| $\epsilon$ + RP + T Cascading | 7108.98 ±0.14 |
| $\epsilon$ + RP + T Linear | 7108.96 ±0.15 |

weighted suboptimal trees. In evolutionary studies, this includes the computation of branch length support (i.e., the probability that a given subtree is present in the true tree), as well as the estimation of confidence intervals for the timing of specific evolutionary events (Drummond et al., 2006). These are particularly important in the very common situations where the data only weakly define the posterior on tree topologies (e.g. small amount of data per species, or high degree of divergence between sequences). In such cases, because suboptimal trees vastly outnumber optimal trees, they can contribute to a non-negligible extent to the probability mass of specific marginal probabilities. Full modeling of posterior distributions on trees is also critical in studying tumor or virus evolution within a patient (McGranahan et al., 2015; Buendia et al., 2009), e.g., to properly assess the probability of the ordered sequence of driver or passenger mutations that may have led to metastasis or drug resistance (Fisher et al., 2013).

## J    REPRODUCIBILITY

In this section, we describe the procedure to reproduce our results. Our most recent implementation involving continuous branch length modeling can be accessed through the following link: `https://github.com/zmy1116/phylogfn/`. For parsimony analysis and bayesian inference with discrete modeling, refer to code and data provided in the supplementary material.

### J.1    MRBAYES

To reproduce the exact MLL results from (Zhang & Matsen IV, 2018a), MrBayes needs to be installed with *Anaconda*.

```
conda install bioconda::mrbayes
```

With MrBayes built from source, the resulted MLL scores differs slightly, as pointed out in (Mimori & Hamada, 2023). We use the following script to compute MLL for DS1–DS8.

```
execute INPUT_FOLDER/ds1.nex
lset nst=1
prset statefreqpr=fixed(equal)
prset brlenspr=unconstrained:exp(10.0)
ss ngen=10000000 nruns=10 nchains=4 printfreq=1000 \
samplefreq=100 savebrlens=yes filename=OUTPUT_FOLDER
```

### J.2    VBPI-GNN

We use the script provided in the paper's GitHub page to generate models for DS1–DS8:

```
python main.py --dataset DS1  --brlen_model gnn --gnn_type edge \
--hL 2 --hdim 100 --maxIter 400000 --empFreq --psp
```

VBPI-GNN requires pre-generation of trees. In the original work, the trees topologies are generated from the algorithm UFBoot (Minh et al., 2013; Hoang et al., 2018). We use the program iqtree (Minh et al., 2020) to run UFBoot on all datasets with following command:

```
iqtree -s DS1 -bb 10000 -wbt -m JC69 -redo
```

To generate randomly perturbed trees to compare log sampling density versus log posterior density. Run the following notebook in our supplementary data folder:

```
vbpi_gnn/generate_trees_data.ipynb
```

### J.3    GEOPHY

We run the following script to run GeoPhy on data set DS1. The script is created based on the best configuration recorded in Mimori & Hamada (2023)

```
python ./scripts/run_gp.py  -v -ip ds-data/ds1/DS1.nex \
-op results/ds1/ds1_paper -c ./config/default.yaml -s 0 -ua \
-es lorentz -edt full -ed 4 -eqs 1e-1 -ast 100_1000 -mcs 3 \
-ci 1000 -ms 1000_000 -ul
```

### J.4    PHYLOGFN-BAYESIAN

Training configuration files for DS1–DS8 are in the following folder in the supplementary data:

```
phylo_gfn/src/configs/benchmark_datasets/bayesian
```

Run the following script to train the PhyloGFN-Bayesian model:

```
cd phylo_gfn
```

```
python run.py configs/benchmark_datasets/bayesian/ds1.yaml \
dataset/benchmark_datasets/DS1.pickle $output_path --nb_device 2 \
--acc_gen_sampling --random_sampling_batch_size 1000 --quiet
```

To compute log MLL, plot sampled log density versus log posterior density and compute the Pearson correlation, run the following notebook in supplementary data:

```
phylo_gfn/phylogfn_bayesian_result.ipynb
```

Models for repeated experiments of full training and short training are trained with a memory efficient PhloGFN implementation in folder **phylo_gfn_bayesian_memoryefficient**

The full training scripts are in folder

```
src/configs/benchmark_datasets/common_script_normal
```

The short training scripts are in folder

```
src/configs/benchmark_datasets/common_script_short
```

Training scripts for ablation study are in folder

```
src/configs/benchmark_datasets/ds1_ablation_study
```

### J.5 PHYLOGFN-PARSIMONY

Training configuration files for DS1–DS8 are in the following folder in the supplementary data:

```
phylo_gfn/src/configs/benchmark_datasets/parsimony
```

Run the following script to train the PhyloGFN-Parsimony model:

```
cd phylo_gfn

python run.py src/configs/benchmark_datasets/parsimony/ds1.yaml \
dataset/benchmark_datasets/DS1.pickle $output_path --nb_device 2 \
--acc_gen_sampling --random_sampling_batch_size 1000 --quiet
```

To generate the plots in Fig. S2, run the following script:

```
cd phylo_gfn
python parsimony_eval.py <model_folder_path> <sequences_path>
```

### J.6 TEMPERATURE-CONDITIONED PHYLOGFN

To reproduce the result in Fig. 3, first run the following script to train the model:

```
cd phylo_gfn

python run.py configs/benchmark_datasets/parsimony/ds1_conditional.yaml \
dataset/benchmark_datasets/DS1.pickle $output_path --nb_device 2 \
--acc_gen_sampling --random_sampling_batch_size 1000 --quiet
```

To generate the plots, run the following notebook in the supplementary data:

```
phylo_gfn/ds1_conditional.ipynb
```

# K   ASSESSING SAMPLING DENSITY AGAINST UNNORMALIZED POSTERIOR DENSITY

Here, we assess PhyloGFN-Bayesian's ability to model the entire phylogenetic tree space by comparing sampling density against unnormalized posterior density over high/medium/low-probability regions of the tree space. To generate trees from medium and low probability regions, we use a trained VBPI-GNN model, the state-of-the-art variational inference algorithm.

We first provide a brief introduction to how VBPI-GNN generates a phylogenetic tree in two phases: (i) it first uses SBN to sample tree topology; (ii) followed by a GNN-based model to sample edge lengths over the tree topology. SBN constructs tree topologies in a sequential manner. Starting from the root node and a set of sequences $Y$ to be labeled at the leaves, the algorithm iteratively generates the child nodes by splitting and distributing the sequence set among the child nodes. The action at each step is thus to choose how to split a set of sequences into two subsets.

To introduce random perturbations during tree construction, at every step, with probability $\epsilon$, the action is uniformly sampled from the support choices. Given a well-trained VBPI-GNN model, we sample from high/medium/low-probability regions with 0%, 30% and 50% probability. We apply PhyloGFN-Bayesian and VBPI-GNN on these sets to compute sampling density and compare with the ground truth unnormalized posterior density computed as $P(Y|z,b)P(z,b)$. Note that VBPI-GNN models the *log-edge length* instead of the edge length. Hence we perform a change of variables when computing both the sampling probability and the unnormalized prior.

Fig. S1 shows scatter plots of sampling density versus ground-truth unnormalized posterior density of datasets DS2 to DS8, complementing the result on DS1 shown in Fig. 2. We can see that while VBPI-GNN performs well on high-probability regions, our method is better at modeling the target distribution overall. Fig. S1f shows that our model performs poorly at modeling the high-probability region for DS7. The result aligns with our MLL estimation, shown in Table S4. Further work is required to investigate the cause of PhyloGFN's poor modeling on DS7.

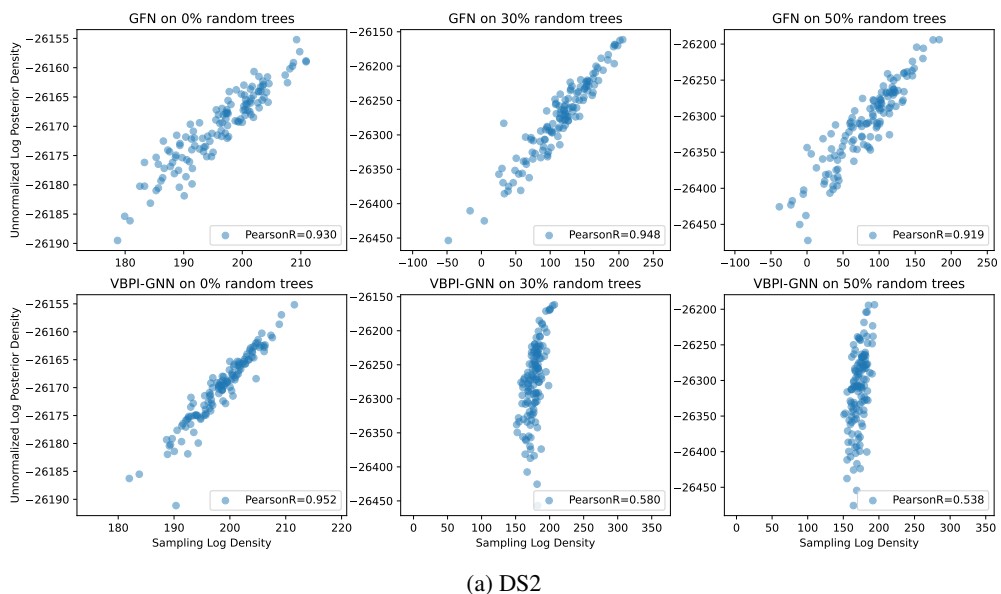

(a) DS2

Figure S1: Sampling log density vs. ground truth unnormalized posterior log density for DS2-DS8

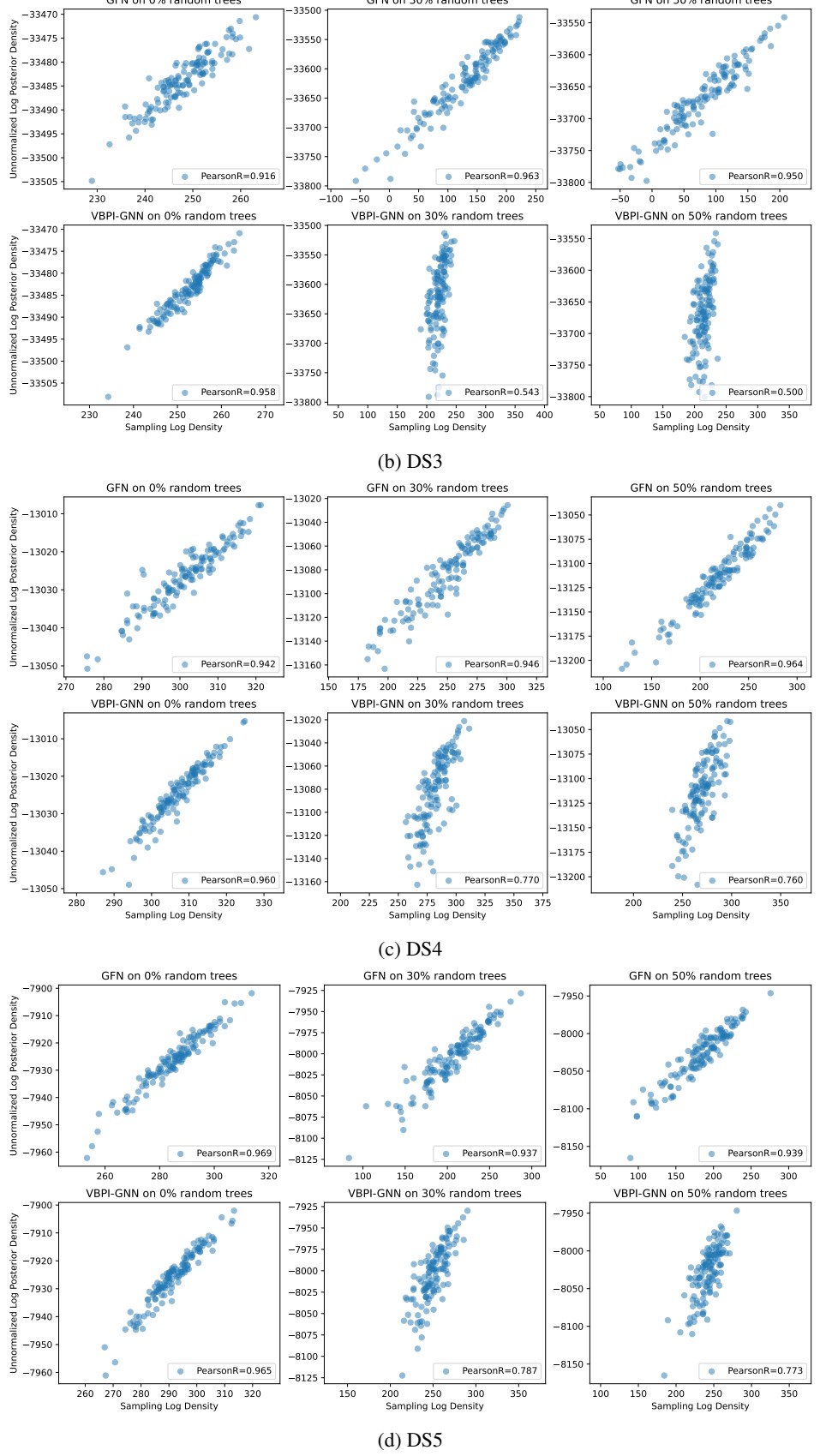

(b) DS3

(c) DS4

(d) DS5

Figure S1: (cont.)

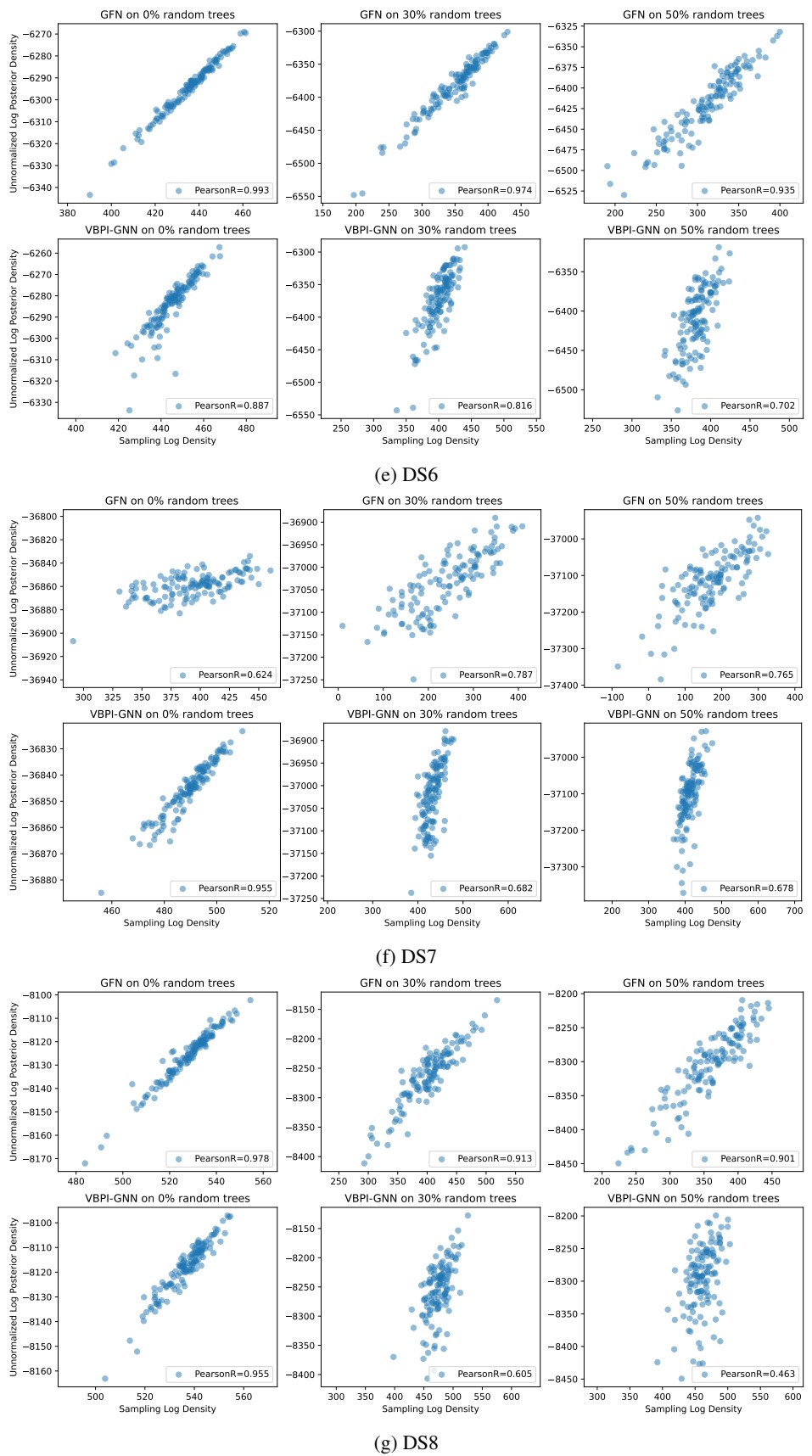

(e) DS6

(f) DS7

(g) DS8

## L  Parsimony analysis results

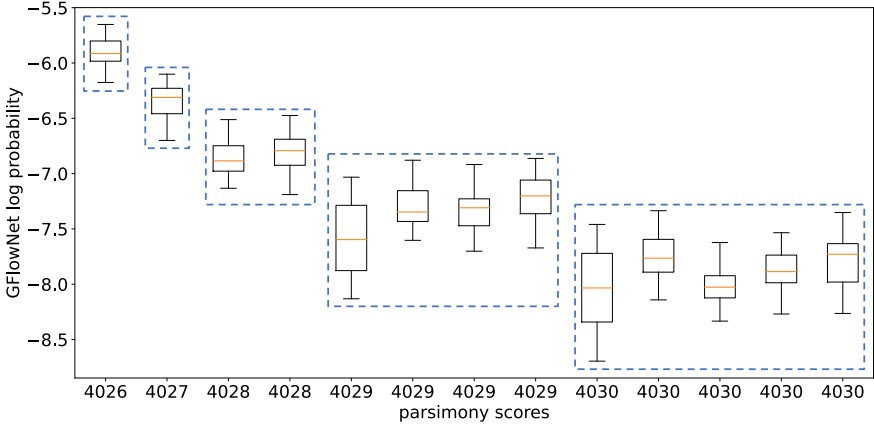

(a) A single most-parsimonious tree with a score of 4026 has been identified for DS1.

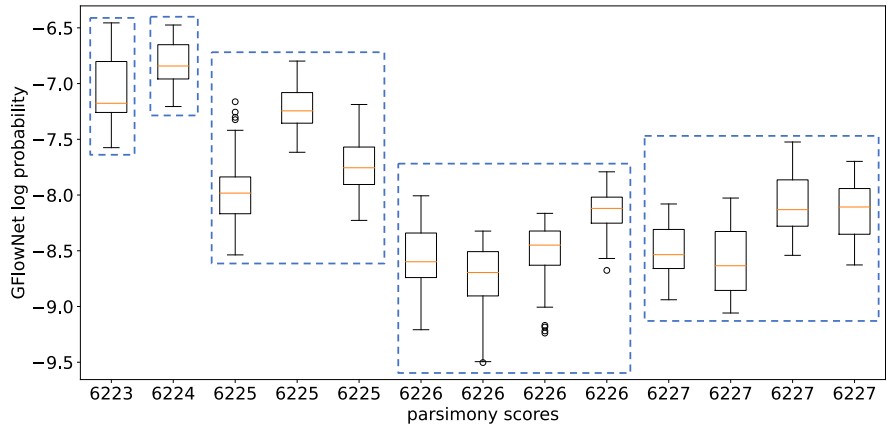

(b) A single most-parsimonious tree with a score of 6223 has been identified for DS2.

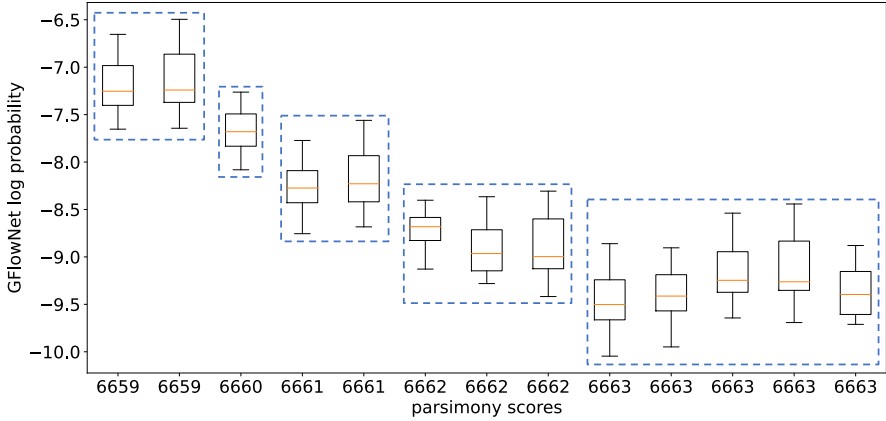

(c) Two most-parsimonious trees with a score of 6659 have been identified for DS3.

Figure S2: For each dataset, the sampling probabilities for the most-parsimonious as well as several suboptimal trees are estimated from our learned PhyloGFNs. Each boxplot represents a single unique unrooted tree and shows a distribution of the log-probabilities over all its $2n - 3$ rooted versions where $n$ denotes the number of species. The dashed bounding box groups the set of equally-parsimonious trees.

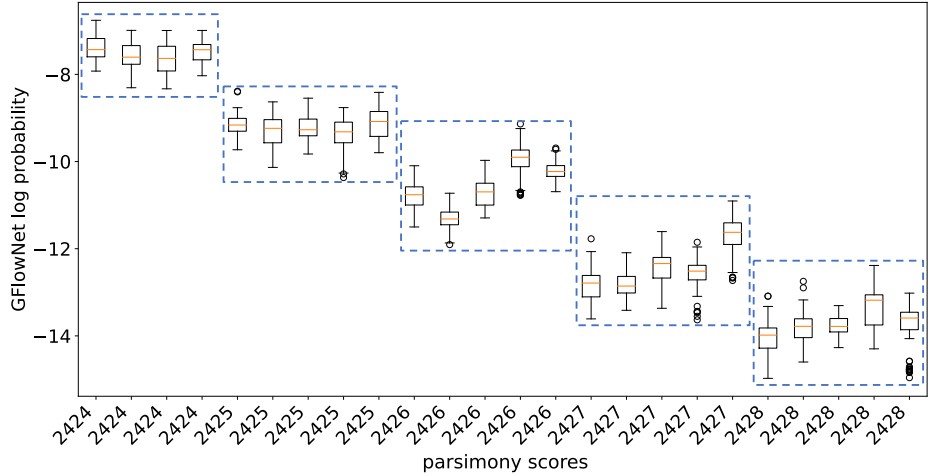

(d) Four most-parsimonious trees with a score of 2424 have been identified for DS4.

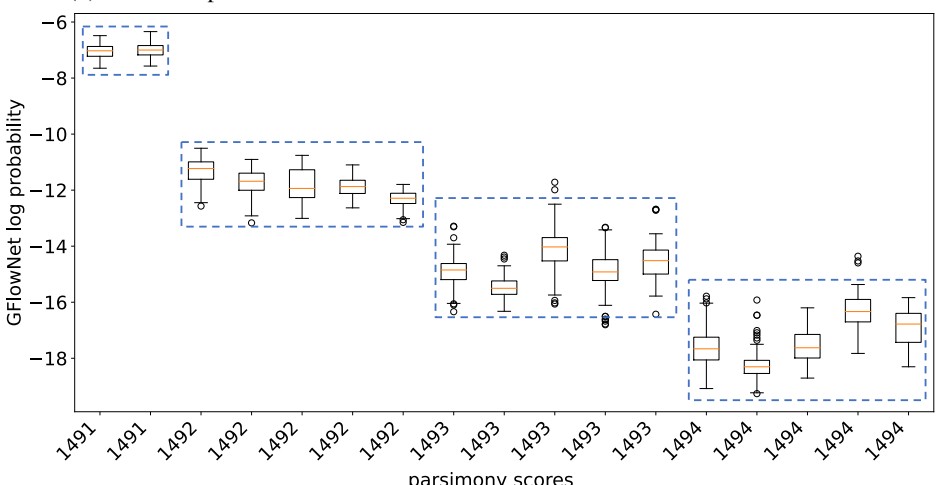

(e) Two most-parsimonious trees with a score of 1491 have been identified for DS5.

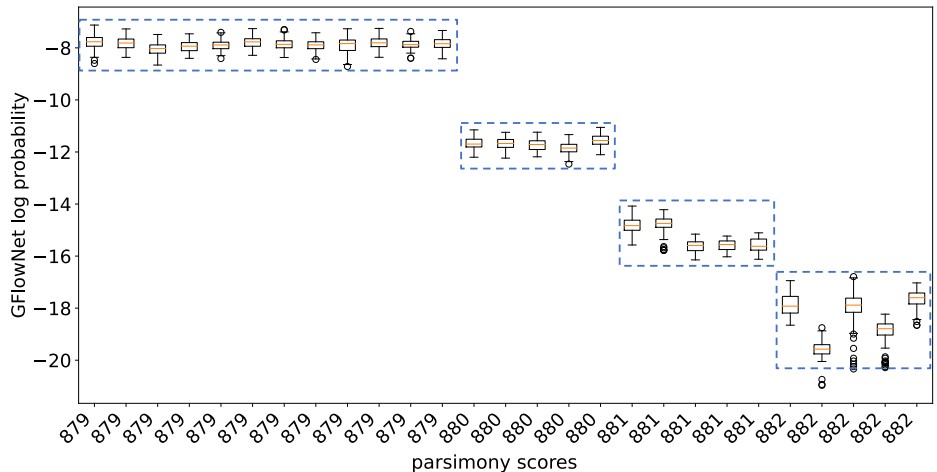

(f) 12 most-parsimonious trees with a score of 879 have been identified for DS6.

Figure S2: (cont.)

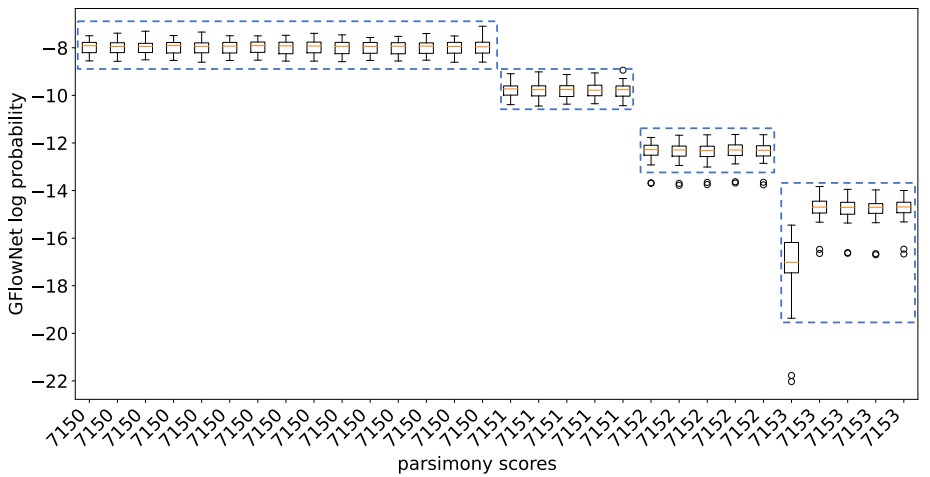

(g) 15 most-parsimonious trees with a score of 7150 have been identified for DS7.

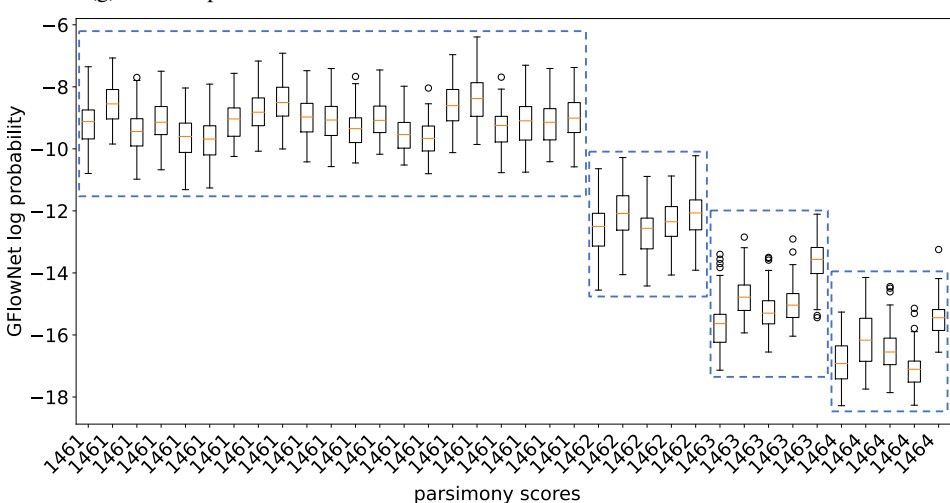

(h) 21 most-parsimonious trees with a score of 1461 have been identified for DS8.

Figure S2: (cont.)

# M    TRAINING CURVES

The plot in Figure S3 illustrates the training curves for PhyloGFN-Bayesian across all eight datasets. In each dataset plot, the left side displays the training loss for every batch of 64 examples, while the right side shows the MLL computed for every 1.28 million training examples.

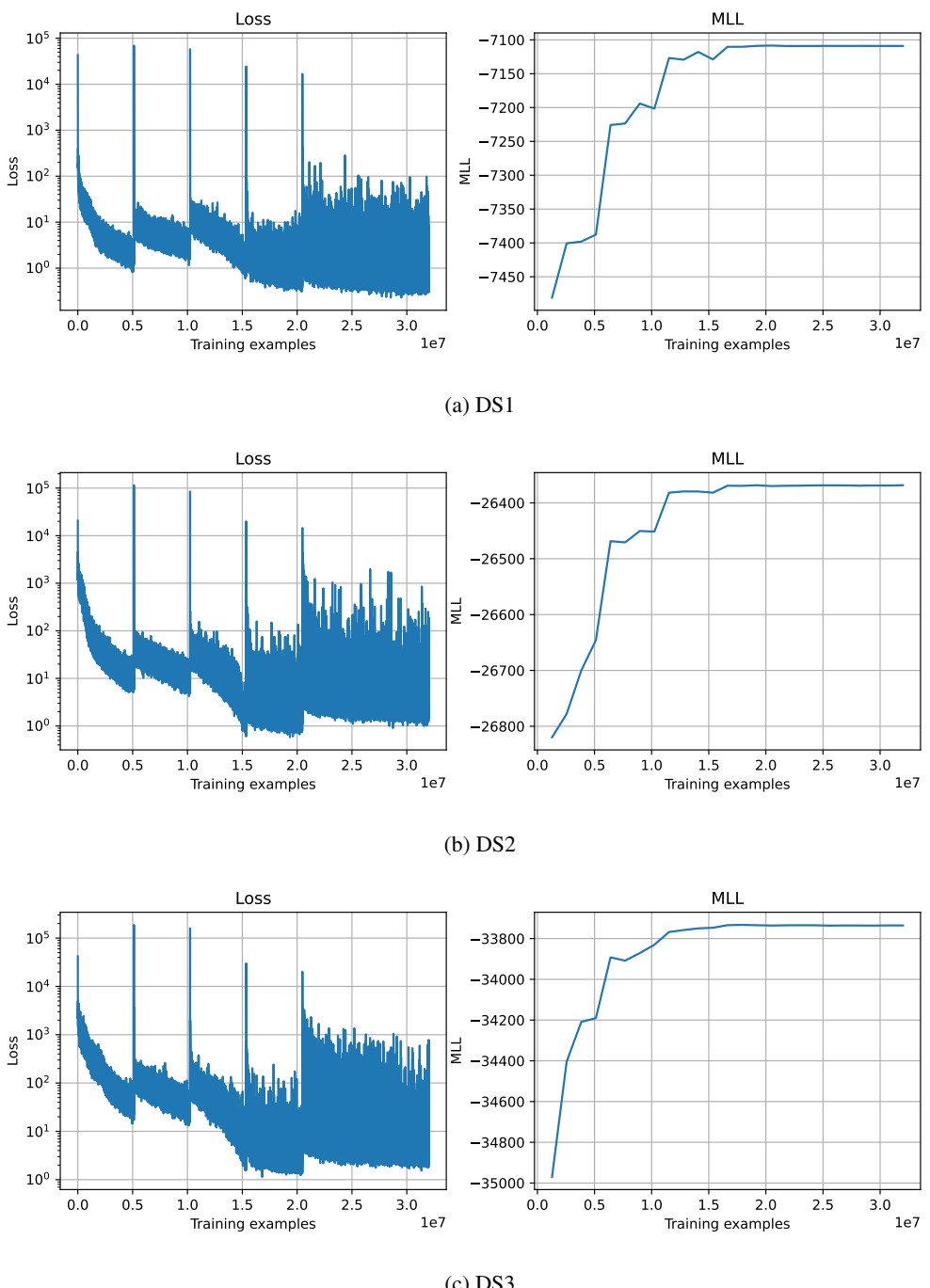

(a) DS1

(b) DS2

(c) DS3

Figure S3: Training curves for DS1-DS8

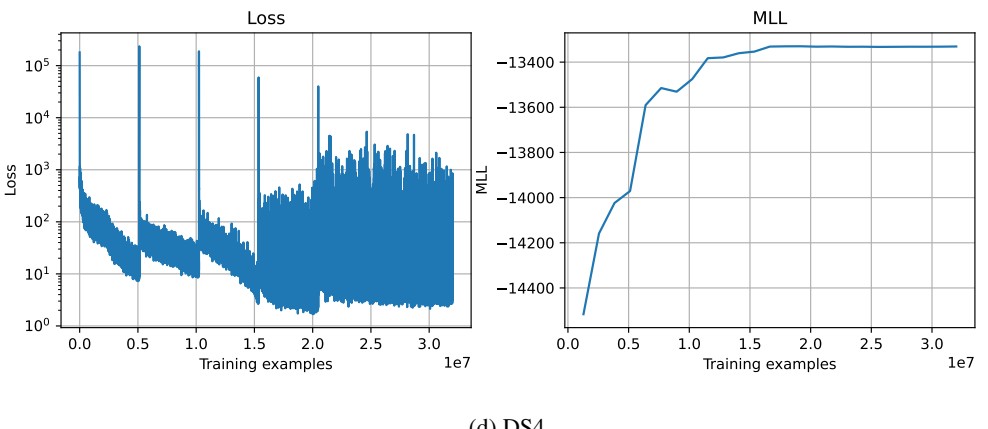

(d) DS4

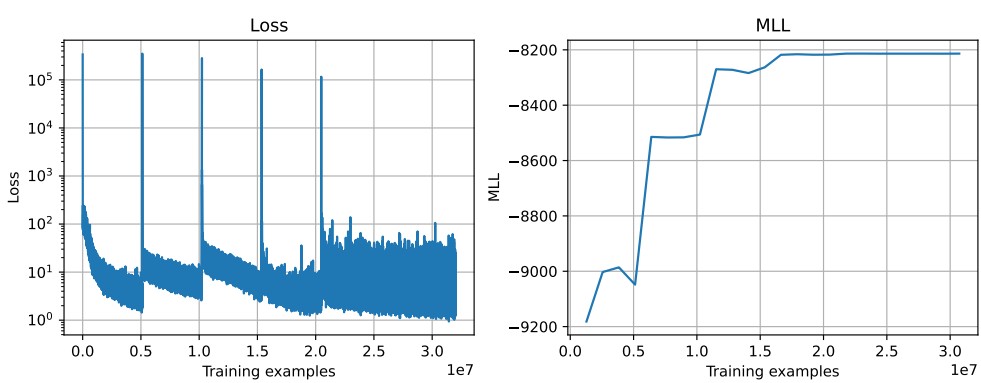

(e) DS5

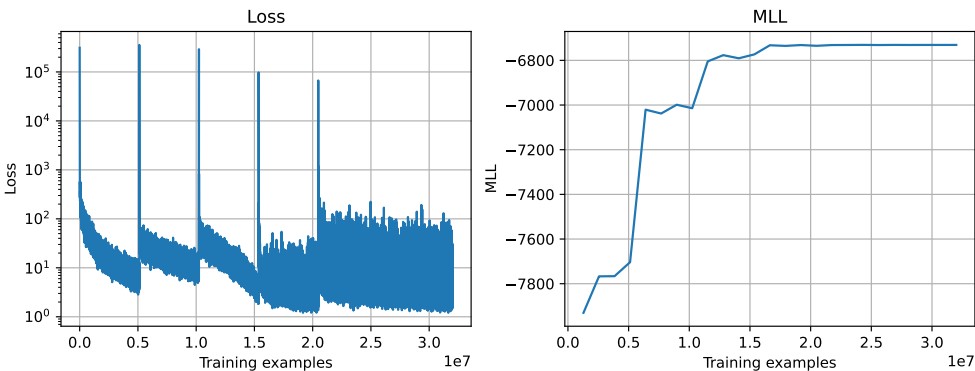

(f) DS6

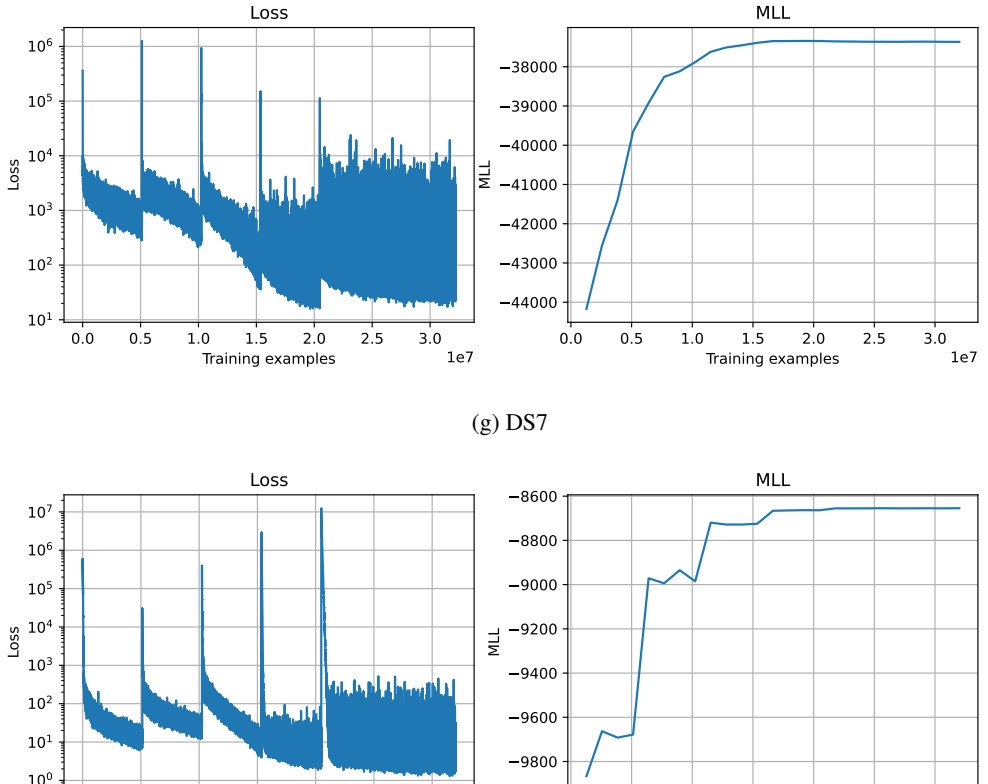

(g) DS7

(h) DS8

