# OpenReview forum: "PhyloGFN: Phylogenetic inference with generative flow networks"
_ICLR.cc/2024/Conference — ICLR 2024 poster_

### Official Review · Reviewer_7zzf · 2023-10-29

**Soundness:** 3 good
**Presentation:** 3 good
**Contribution:** 3 good
**Rating:** 6
**Confidence:** 4

**Summary:**

This is a nice application of GFlowNets to the phylogeny problem in computational biology. There is a main issue that needs to be resolved, namely the performance. First the marginal likelihood comparison with other methods, are the PhyloGFN better in terms of lower bound and what is the running time required to get these results. Second, regarding the better estimates of posterior probabilities for low posterior  phylogenetic tree (which may be the main selling point for this paper), what is the gold standard and how does your running time compare with that of he gold standard and those of the other methods? Running time doesn't have to be measured as wall-clock or using a formal analysis, but could be based on the number of so-called Felsenstein peeling operations (or whatever that make it likely that this approach may with sufficient effort yield a efficient method in the future, if that isn't already the case). The proper resolution of these issues should make this become a good contribution.

**Strengths:**

The strength is really the novelty and that this method both obtains good performance and is not limited to a predefined set of trees.

**Weaknesses:**

It is not really trying to explain the GFN approach to the reader, but rather enumerating required items. The running time and performance in general require further explanation. The parsimony case is clearly less interesting and could partially be move to the appendix.

**Questions:**

high complexity of tree space

■ Page 1
It is large! complex is less clear.




PhyloGFN is competitive with prior works in marginal likelihood estimation and achieves a closer fit to the target distribution than state-of-the-art variational inference methods

■ Page 1



I don’t know, but the better fit for lower part of the posterior is worth mentioning. It may be key.





continuous vari- ables that capture the level of sequence divergence along each branch of the tree.

■ Page 1



Also the ml version has those.





Coming from the intersection of variational inference and reinforcement learning is

■ Page 1



Reformulate.





PhyloGFN leverages a novel tree representation inspired by Fitch and Felsenstein’s algorithms to represent rooted trees without introducing additional learnable parameters

■ Page 2



Make this clearer. In particular, in addition to what?





explores

■ Page 2



No it has capacity or potential to do this.





state-of-the-art MCMC

■ Page 2



The comparison must be made relative to resources since you use mcmc as gold standard





With their theoretical foundations laid out in Bengio et al. (2023);

■ Page 2



Is this the gfn sota? Then point out this fact.





The tree topology can be either a rooted binary tree or a bifurcating unrooted tree.

■ Page 2



This is a potential weakness. Can you restrict you method to binary trees? Does this mean that the posterior support of a subsplit consist of both binary and multifurcating trees? On an earlier reading i got the impression that you considered binary trees. Please clarify.





This equation can be used to recursively compute 𝑃(𝐿𝑖 𝑢|𝑎𝑖 𝑢) at all nodes of the tree and sites 𝑖. The algorithm performs a post-order traversal of the tree,

■ Page 3



This is called dynamic programming





recursively computed by (1)

■ Page 3



Again DP





The algorithm traverses the tree two times, first in post-order (bottom-up) to calculate the character set at each node, then in pre- order (top-down) to assign optimal sequences.

■ Page 4



Also this is DP





Generative flow networks (GFlowNets) are algorithms for learning generative models of complex distributions given by unnormalized density functions over structured spaces. Here, we give a con- cise summary of the the GFlowNet framework.

■ Page 4



This section could provide more insight





The direct optimization of 𝑃𝐹’s parameters 𝜃 is impossible since it involves an intractable sum over all complete trajectories.

■ Page 4



This argument is incorrect. With polynomial length trajectories, which you need, exponentially many trajectories implies exponentially many states. The ladder also yields your optimization infeasible, in worst case.





Each action chooses a pair of trees and join them at the root, thus creating a new tree

■ Page 5



This suggests binary trees. “At the root” is a poor formulation.





hus, a state 𝑠 consists of a set of rooted trees

■ Page 5



Disjoint rooted …





its two children

■ Page 5



This should close the case!





all of which can be reached by our sampler

■ Page 5



Are all probabilities always non zero ?





State representation To represent a rooted tree in a non-terminal state, we compute features for each site independently by taking advantage of the Felsenstein features (§3.1.1).

■ Page 5



Can you motivate this choice?





Although the proposed feature representation 𝜌 does not capture all the information of tree structure and leaf sequences, we show that 𝜌 indeed contains sufficient informa- tion to express the optimal policy

■ Page 6



What is the optimal policy? How do you show this?





Proposition 1. Let 𝑠1 = {(𝑧1, 𝑏1), (𝑧2, 𝑏2) . . . (𝑧𝑙, 𝑏𝑙)} and 𝑠2 = {(𝑧′ 1, 𝑏′ 1), (𝑧′ 2, 𝑏2) . . . (𝑧′ 𝑙, 𝑏𝑙)} be two non-terminal states such that 𝑠1 ≠ 𝑠2 but sharing the same features 𝜌𝑖 = 𝜌′ 𝑖. Let 𝒂 be any sequence of actions, which applied to 𝑠1 and 𝑠2, respectively, results in full weighted trees 𝑥 = (𝑧, 𝑏𝑧 ), 𝑥′ = (𝑧′, 𝑏′), with two partial trajectories 𝜏 = (𝑠1 → · · · → 𝑥), 𝜏′ = (𝑠2 → · · · → 𝑥′). If 𝑃𝐹 is the policy of an optimal GFlowNet with uniform 𝑃𝐵, then 𝑃𝐹 (𝜏) = 𝑃𝐹 (𝜏′

■ Page 6



Please explain the importance.





−7108.42 ±0.18 −7108.41 ±0.14 −7290.36 ±7.23 −7111.55 ±0.07 −7108.95 ±0.06

■ Page 7



At least the second method is a lower bound and it entire interval os above yours. Isn’t that better?





Pearson correlation of model sampling log-density and ground truth unnormalized posterior log-density for each dataset o

■ Page 8



How is the ground truth obtained?





589

■ Page 8





512

■ Page 8





0.624

■ Page 8



Fails miserably





BAYESIAN PHYLOGENETIC INFERENCE

■ Page 8



What is the computational resources required for these methods?





The base- lines we compare to are the MCMC-based MrBayes combined with the stepping-stone sampling tech- nique (Ronquist et al., 2012),

■ Page 8



How do you obtain that and are you sure of how it is computed?





To select pairs of trees to join, we evaluate tree-pair features for every pair of trees in the state and pass these tree-pair features as input to the tree MLP to generate probability logits for all pairs of trees. The tree-pair feature for a tree pair (𝑖, 𝑗) with representations 𝑒𝑖, 𝑠𝑗 is the concatenation of 𝑒𝑖 + 𝑒 𝑗 with the summary embedding of the state, i.e., the feature is [𝑒𝑠; 𝑒𝑖 + 𝑒 𝑗], where [·; ·] denotes vector direct sum (concatenation). For a state with 𝑙 trees, 𝑙 2 = 𝑙(𝑙−1) 2 such pairwise features are generated for all possible pairs

■ Page 16

---

> ### Author Response · Authors · 2023-11-17
> **Response for reviewer 7zzf**
>
> We appreciate your feedback and detailed comments. We may not have fully comprehended some of your questions. If this is the case, please let us know, and we will be happy to clarify.
>
> ## GFlowNets introduction
> > It is not really trying to explain the GFN approach to the reader, but rather enumerating required items
> >
> >This section could provide more insight (background section on GFlowNets)
>
> Space limitations prevent us from giving a complete introduction to GFlowNets, but we believe that our introduction is self-contained and sufficient for understanding the application to phylogenetic inference. However, we have tried to improve the presentation. We point the reader to ["Trajectory balance...", NeurIPS 2022] as an accessible and self-contained introduction to the GFlowNet framework as relevant to our setting.
>
> ## Running time and computation resources
> > The running time and performance in general require further explanation
> >
> > The comparison must be made relative to resources since you use mcmc as gold standard
> >
> >What is the computational resources required for these methods?
>
> We describe the computation resources for PhyloGFN and compare the running time with other benchmark algorithms in our answer to all reviewers. We hope this answers your question.
>
> ## Move the parsimony analysis to appendix
>
> > The parsimony case is clearly less interesting and could partially be move to the appendix
>
> Bayesian inference and parsimony analysis are two types of problems that are usually solved by different families of algorithms. We believe that the fact that PhyloGFN is robust enough to solve both of them under the same framework is an important contribution worth highlighting. In particular, the following technical contributions related to parsimony-based inference are noteworthy:
>
> - We solve the parsimony score minimization problem with a sampling approach by designing an energy distribution with the parsimony score as the energy. With sufficiently small temperature $T$, the distribution will be dominated by optimal solutions, and the learned GFlowNet can retrieve all parsimonious trees.
> - We propose a tree representation based on Fitch node features. While it has similarity to the Felsenstein-based tree features for Bayesian inference, the proof of its sufficiency for optimal policy is different.
> - We propose a temperature-conditioned PhyloGFN that takes $T$ as input and can thus trade off between diversity of tree topologies and low parsimony score at inference time.
>
> ## GFlowNet SOTA
> > With their theoretical foundations laid out in Bengio et al. (2023); (Page 5)
> >
> > Is this the gfn sota? Then point out this fact.
>
> We are not sure what you mean by this question. GFlowNets do indeed achieve state-of-the-art sampling performance for many problems, but they never been applied to phylogenetic inference. The adaptation of GFlowNets to phylogenetic inference is a major contribution of our work.
>
> ## On multifurcating trees
> > The tree topology can be either a rooted binary tree or a bifurcating unrooted tree. (Page 2)
> >
> > This is a potential weakness. Can you restrict you method to binary trees? Does this mean that the posterior support of a subsplit consist of both binary and multifurcating trees? On an earlier reading i got the impression that you considered binary trees. Please clarify.
>
> We believe there is a misunderstanding here. Like almost all approaches in phylogenetics (including all the recent work addressing the problem with machine learning), we do not consider multifurcating trees. The two types of trees considered are rooted and unrooted binary trees.
>
> ## Question on trajectories/states of GFlowNets
>
> > With polynomial length trajectories, which you need, exponentially many trajectories implies exponentially many states. The ladder also yields your optimization infeasible, in worst case.
>
> We respectfully disagree that this is a limitation. All machine learning-based phylogenetic inference algorithms learn to sample a distribution over an exponentially large space of of tree topologies, but they use the power of amortization and deep neural nets to generalize from a much smaller number of topologies seen in training.
>
> To be precise, given $n$ sequences, there are indeed $(2n-5)!!$ tree topologies, which is superexponential. You are correct that it is unrealistic to let the model visit every single tree to learn the full distribution. This is exactly the reason we use deep neural networks to parameterize the action policy. With deep neural networks, the algorithm can learn the underlying statistical patterns of the trees and generalize to those unseen in training.

---

> > ### Author Response · Authors · 2023-11-17
> >
> > ## Sampling probabilities of trees
> > > all of which can be reached by our sampler (Page 5)
> > >
> > > Are all probabilities always non zero ?
> >
> > Indeed, as the sampling probability from any state to any of its child states is nonzero, for every tree that can be constructed with our defined construction procedure, the sampling probability is always strictly positive (although it can be quite small).
> >
> > ## Intuition on our tree representation for Bayesian inference
> > > State representation To represent a rooted tree in a non-terminal state, we compute features for each site independently by taking advantage of the Felsenstein features (page 5)
> > >
> > > Can you motivate this choice? (of designing tree representation based on Felsenstein's algorithm)
> >
> > Absolutely. Since our PhyloGFN’s policy model takes each tree's representation as input, the representation needs to provide sufficient information for the policy to sample actions. The policy model is trained with the learning objective that sampling probability is proportional to the unnormalized posterior probability. Therefore, the feature set should contain sufficient information for the model to compute the unnormalized posterior probability $P(z,b,Y)$.
> >
> > This measure can be calculated using Felsenstein’s algorithm in a bottom-up fashion through dynamic programming (see Section 2). At every internal node we calculate and store the conditional likelihood of character assignments given the observations, and we refer to these as Felsenstein features. In Proposition 1, we prove that using Felsenstein features as node representations gives sufficient information for modeling the optimal sampling policy.
> >
> > ## Question on tree feature representation power
> > > What is the optimal policy? How do you show this?
> >
> > A policy is optimal when its sampling probability of a tree $(z,b)$ equals to the ground truth posterior probability of the tree $P(z,b|Y)$. This is equivalent to the policy having having 0 trajectory balance loss for every trajectory in the Markov decision process.
> >
> > > Please explain the importance. (of proposition 1)
> >
> > We are aiming to show that our tree representation contains enough information for an optimal policy to make decisions. The formal statement of this result is that "in an optimal policy, two states with identical features share the same transition probabilities", which is what we show in Proposition 1. Please let us know if this has helped answer the question.
> >
> > ## VBPI-GNN has higher MLL estimation performance
> > > At least the second method is a lower bound and it entire interval os above yours. Isn’t that better? (Page 7 Table 1 MLL results of DS1)
> >
> > Indeed, the estimated MLL of VBPI-GNN is higher than that of PhyloGFN on DS1. As acknowledged in the results section, we have highlighted VBPI-GNN's superior MLL for benchmark datasets in our paper, attributing it to the requirement of a set of pre-defined tree topologies likely to achieve high likelihood -- a form of biological prior knowledge. However, this advantage comes with a significant drawback: VBPI-GNN's performance is constrained by the pre-defined set of trees. And these pre-defined trees also limits VBPI-GNN’s sample space to only a portion of phylogenetic tree space. Moreover, as illustrated in Figure 2, Figure S1, and Table 2, VBPI-GNN does not model suboptimal trees well.
> >
> > It is also important to note that PhyloGFN outperforms other VI methods, such as GeoPhy and VaiPhy, that are capable of modeling the entire phylogenetic tree space. Additionally, PhyloGFN demonstrates a superior ability to fit suboptimal trees compared to VBPI-GNN.
> >
> > Please also see the response to all reviewers regarding the importance of modeling the full tree space.
> >
> > ## ground truth for analysis in figure 2 and table 2
> > > Pearson correlation of model sampling log-density and ground truth unnormalized posterior log-density for each dataset o (Page 8)
> > >
> > > How is the ground truth obtained?
> >
> > The ground truth is the logarithm of the unnormalized posterior density $log( P(Y|z,b)P(b)P(z))$, where $P(Y|z,b)$ can be calculated using Felsenstein’s algorithm, $P(z)$ is $1/ (2n-5)!!$ by the uniform prior assumption, and $P(b) = \prod_e P(b(e)), P(b(e)) = \text{Exp}(10) (b(e))$.
> >
> > ## Under-performing results
> >
> > > 589,512, 0.624, Fails miserably
> >
> > By "589" and "512", we assume this reviewer is referring to VBPI-GNN's PearsonR score for DS1 suboptimal trees. Indeed, our analysis in Table 2 and Figure 2 shows that PhyloGFN is superior at modeling suboptimal trees compared to VBPI-GNN.
> >
> > By "0.624", we assume this reviewer is referring to the PhyloGFN PearsonR score for DS7. Yes, PhyloGFN models poorly the high-probability trees on this particular dataset.

---

> ### Author Response · Authors · 2023-11-17
>
> ## Mr Bayes benchmark
> > The base- lines we compare to are the MCMC-based MrBayes combined with the stepping-stone sampling tech- nique (Ronquist et al., 2012) (page 8)
> >
> > How do you obtain that and are you sure of how it is computed?
>
> Thank you for pointing out that our description for this benchmark result is unclear. Stepping stone is the MCMC-based sampling algorithm to generate phylogenetic trees and estimate MLL scores. MrBayes is the software that implements the stepping stone algorithm.  In table 1, the MR-Bayes results for MLL estimation are obtained by using MrBayes to run the Stepping stone algorithm.  We will update the manuscript to resolve this confusion.
>
> In table 1, the displayed results are the MrBayes results recorded in GeoPhy, VBPI-GNN.  To perform the computation, they follow the same procedure described in  Zhang \& Matsen IV (2018b). In appendix section F1, we provide the MrBayes script used in Zhang \& Matsen IV (2018b).
>
> ## Question on model setup
>
> > To select pairs of trees to join, we evaluate tree-pair features for every pair of trees ... (Page 16)
>
> We didn't see a question or comment for the last section in your questions. Please let us know if you have any question on the model setup.
>
> ## Question about continuous branch length variables
>
> > continuous vari- ables that capture the level of sequence divergence along each branch of the tree. (Page 1)
> >
> >Also the ml version has those.
>
> We are not sure what you mean by this comment. We are happy to answer if you could clarify the question.
>
>
> ## On rooted/unrooted trees
>
> > Each action chooses a pair of trees and join them at the root, thus creating a new tree (page 5)
> >
> > This suggests binary trees.  “At the root” is a poor formulation.
> >
> > its two children (page 5)
> >
> > This should close the case!
>
> We are not quite sure what you mean here, but we can revise the sentence to end "each action chooses a pair of trees and joins their roots by a common parent node".
>
> ## Suggestions for edits
>
> > high complexity of tree space (Page 1)
> >
> > It is large! complex is less clear.
>
> Thank you for pointing out this issue.  We will make the change “high complexity of tree space” to “extremely large tree space” in the manuscript.
>
> > I don’t know, but the better fit for lower part of the posterior is worth mentioning. It may be key.
>
> We agree with you. We will update the manuscript to highlight the significance of modeling suboptimal trees.
>
> > Coming from the intersection of variational inference and reinforcement learning is
> > Reformulate.
>
> We will reformulate this to make it more clear.
>
> > This is called dynamic programming (for Felsenstein's algorithm)
>
> Yes, the recursive computation in Felsenstein's algorithm is an instance of dynamic programming. We can use the term "dynamic programming" if the text if you find it suitable.
>
> >a state $s$ consists of a set of rooted trees
> >
> >  Disjoint rooted …
>
> Indeed, $s$ is a disjoint set of rooted trees (that is, a forest). We will update the manuscript accordingly to include the word "disjoint".
>
> > PhyloGFN explores and samples from the entire phylogenetic tree space, achieving a balance between exploration in this vast space and high-fidelity modeling of the modes. (Page 2)
> >
> > No it has capacity or potential to do this.
>
> Thanks for the suggestion, we will update the manuscript to "PhyloGFN has the capacity to explore ...".
>
> > PhyloGFN leverages a novel tree representation inspired by Fitch and Felsenstein’s algorithms to represent rooted trees without introducing additional learnable parameters (Page 2)
> >
> > Make this clearer. In particular, in addition to what?
>
> Thanks for pointing this out. We will update the manuscript to "without introducing additional learnable parameters to the model" .
>
> We hope we have addressed all of your concerns. Please let us know if something else needs further clarification or if we missed something.

---

> ### Author Response · Authors · 2023-11-21
>
> Dear Reviewer 7zzf,
>
> We have responded to your questions and comments above. Could you please let us know if you have any further questions before the end of the rebuttal period and if they have affected your assessment of the paper?
>
> We have also just posted a revised pdf. Please see the comment to all reviewers for details of the changes.
>
> Thank you,
>
> The authors

---

> > ### Comment · Reviewer_7zzf · 2023-11-21
> > **Mr Bayes and the joining of trees**
> >
> > Thanks for all the clarifications.
> >
> > I still haven’t understood exactly what the potential of the methods is. Can you provide evidence that this method with additional optimization etc will able to compete with MCMC approaches, in particular Mr Bayes. That is, can you get better accuracy or a faster inference, or both? Again, speed doesn’t have to be wall-clock but measured in some relevant way, which at least partially removes the advantage of Mr Bayes, i.e.,  being an optimized mature software.
> >
> > Regarding “To select pairs of trees to join, we evaluate tree-pair features for every pair of trees ... (Page 16)”. The intended question is so: do you have so consider all pairs of roots among the trees? Can you give a more extensive account of the transformer, which is crucial since it allows you to, in contrast to VBPI, potentially can consider any possible tree? Also, what is it’s complextity etc?
> >
> > I find the paper very interesting and look forward to having also these issues clarified.

---

> > > ### Author Response · Authors · 2023-11-22
> > > **Response for reviewer 7zzf**
> > >
> > > Dear reviewer 7zzf,
> > >
> > > Thank you for your questions and clarifications. We address your new questions and comments below.
> > >
> > > ## Comparison with MCMC-based algorithms
> > >
> > > Compared with MCMC algorithms, the advantages of PhyloGFN are 1. PhyloGFN is an amortized sampler, able to very quickly generate a large number of independent tree samples once it has been trained, 2. scalability to high-dimensional distributions with multiple separated modes.
> > >
> > > To compare with MrBayes SS sampling specifically, the result in Table 1 is generated by 10 repetitions of 10 million steps of SS sampling. Our PhyloGFN Short experiment runs for a similar number of samples (trees visited) -- 12.8 million training examples, as described in the answer to all reviewers. Therefore, we compare these two algorithms on the DS1 dataset. Besides the running time and MLL estimation, we also follow the analysis from the official VaiPhy GitHub repository (`vaiphy/benchmarks/mrbayes/mrbayes.py`) to compute aggregate statistics of SS sampling trees. We compute the highest log-likelihood (Max LL) and average log likelihood (Mean LL) of sampled trees and compare them with the highest log-likelihood tree visited by PhyloGFN during training and the average log-likelihood of 1024 sampled trees at inference time. The results are shown in the table below. The std for MLL,  Max LL and Mean LL are computed  over multiple experiments (3 runs for PhyloGFN Short, 10 runs for MRBayes SS).
> > >
> > > | Method                      | Running Time | MLL             | Max LL          | Mean LL         |
> > > |-----------------------------|--------------|-----------------|-----------------|-----------------|
> > > | MrBayes SS (10M steps)     | 9h10min      | -7108.42 (0.18) | -6896.83 (0.92) | -6907.02 (0.44) |
> > > | PhyloGFN Short (12.8 steps) | 20h40min     | -7108.96 (0.07) | -6893.31 (0.27) | -6903.25 (0.27) |
> > >
> > >
> > > In terms of estimating MLL, MrBayes SS outperforms PhyloGFN and all amortized / VI algorithms discussed in this paper (we note that MrBayes is a widely accepted gold standard in Bayesian phylogenetic inference). It is the fastest algorithm, and it produces the state-of-the-art MLL estimation. However, it needs to be re-emphasized that PhyloGFN (and VI algorithms) learns an amortized sampler. **Once the model is trained, it is much faster at sampling new trees than MrBayes.** If we use the same MrBayes SS setup to sample 100 trees that are 1000 MCMC steps apart from each other, it takes roughly 500 seconds.  On the other hand, sampling 100 independent trees from a trained PhyloGFN takes **less than 1 second**.
> > >
> > > Looking at the aggregated statistics of samples, although it appears that PhyloGFN can sample higher-likelihood trees than SS, it is worth noting that SS varies the statistical temperature through the training. Therefore, some low-likelihood trees are included in the SS samples. However, the max LL is a valid metric that indicates PhyloGFN has seen trees with higher LL than MRBayes SS.
> > >
> > > The script to run MrBayes SS sampling is provided in Appendix I.1 in the manuscript.

---

> > > > ### Comment · Reviewer_7zzf · 2023-11-22
> > > > **Not convinced but believe in the paper's potential**
> > > >
> > > > I'm not convinced by the division of the running time into training and sampling. In most situations, it is the overall time that matters. Nevertheless, I believe in the paper's potential and will raise my score.

---

> > > > > ### Author Response · Authors · 2023-11-22
> > > > >
> > > > > Dear reviewer 7zzf,
> > > > >
> > > > > Thank you for taking the effort of reviewing our paper.
> > > > >
> > > > > The authors

---

> ### Author Response · Authors · 2023-11-22
>
> ## Tree topology construction and transformer model
>
> >  do you have so consider all pairs of roots among the trees?
>
> Yes, given a state consisting of $N$ disjoint trees, we need to choose a pair of trees out of all $N \choose 2$ possible pairs. We do not use any heuristic to limit the set of pairs. In this way, PhyloGFN models the entire tree space.
>
>
> > Can you give a more extensive account of the transformer, which is crucial since it allows you to, in contrast to VBPI, potentially can consider any possible tree?
>
> We want to emphasize that the capability of modeling the entire tree space does not depend on the specific architecture of the transformer model. As we explained above, PhyloGFN can sample all possible trees because the action policy covers all possible pairs of trees at every state. In the PhyloGFN architecture, this is achieved by applying an MLP to each pair of condensed tree features, yielding logits for all possible actions of joining two trees by a common parent node. The transformer encoder is utilized to generate condensed feature representations of trees. While a transformer architecture is suitable for our input, which consists of a set of disjoint trees, it is not the determining factor enabling PhyloGFN to model the entire tree space. This is also why we relegated the architecture details to the appendix in our paper.
>
> That being said, we are happy to provide more details regarding the transformer architecture itself: the hyperparameters used for the transformer encoder are recorded in Table S3 in the appendix. For all datasets, the transformer encoder consists of 6 self-attention blocks with 4 heads in each block. The feature embedding size is 128 and the activation function is GELU. Each self-attention block's layout is similar to the self-attention block of the ViT model (Dosovitskiy et al., 2020). It consists of a layer normalization, a self-attention layer, another layer normalization, and finally an MLP block. For a problem with $N$ input sequences, the time complexity of an $L$-layer transformer is $O(N^2 L)$.
>
>
> We hope that our responses have addressed your questions and concerns sufficiently. Please let us know if we can provide any more clarifications.

---

### Official Review · Reviewer_SMRb · 2023-10-30

**Soundness:** 3 good
**Presentation:** 2 fair
**Contribution:** 3 good
**Rating:** 8
**Confidence:** 3

**Summary:**

The paper develops a method for inferring phylogenetic trees, i.e. graphical representations of the evolutionary relationships of species, using Generative Flow Networks (GFlowNets). The GFlowNet treats the tree building processes as a reinforcement learning problem, where the action set corresponds to joining the roots of the subtrees existing in the pool. The paper uses a transformer architecture to encode the input states that correspond to a set of features extracted from subtree structures.

**Strengths:**

* Both the proposed model architecture and the use of GFlowNets for the chosen application are novel and interesting.

 * The paper conducts a large body of well-planned experiments, compares against a properly chosen set of baselines and reports results favorable to the proposed method.

 * The fact that the comparison is not made against many alternative methods is understandable, as there probably are not many modern machine learning methods addressing the same problem.

* The design choices used in the model architecture are well justified, for instance the one given at the end of Page 6 for the transformer makes perfect sense.

**Weaknesses:**

* The biggest weakness looks to me like the results in Table 1. The log-likelihood scores look very similar to each other. For instance -7108.95 for PhyloGFN and 7108.95 VBPI-GNN. Similar for other data sets.

 * It is great that the paper makes lots of effort to ensure the reproducibility of the results. However, It looks to me like the main paper lacks some essential details about the experiments, making it a bit hard for the reader to evaluate the results. See my question below.

 * Likewise, the paper would be more readable by the machine learning community if the used tree-level features are explained. As far as I was able to detect, the main paper mentions only that they are Fitch and Felsenstein features, which may be obvious to an evolutionary biologist but they do not tell anything to me as a machine learning researcher. Now that this is not a biology journal but a machine learning research venue, an introduction to such basic concepts somewhere in the paper could be beneficial.

 * The paper reports log-likelihood results, which measures model fit. It measures parsimony, which appears to be about the computational aspect. The log densities reported in Figure 2 is an indicator of diversity. It may be better to have a more direct and interpretable score of the discovery performance, for instance prediction accuracy of evolution tree links or ancestral relationship detection between pairs of species in cases where there is agreement on the ground truth. The current result landscape looks a bit too exploratory. While the results make intuitive sense, they still leave many gray areas in their detailed interpretation.

**Questions:**

Are the \pm values given in Table 1 standard deviation or standard error? What are these standard deviations/error over? Experiment repetitions? If standard deviation across repetitions, the results may be good enough. They may be alarming otherwise.

Overall this is an interesting piece of work with decent potential for impact. I also give sincere value to the effort the authors make to use advanced machine learning methods for such hard applications of natural sciences. However, the paper requires some work to improve, especially in terms of presentation and clarification before being ready for publication. The case looks to me like borderline at present but has potential to improve towards an accept after the rebuttal.

---

The rebuttal has addressed my concerns, so I raise my score to an accept.

---

> ### Author Response · Authors · 2023-11-17
>
> Thank you for your feedback and interesting questions. We address them in turn below.
>
> ## MLL estimation similar among different algorithms
>
> While VBPI-GNN demonstrates MLL performance closely aligned with the state-of-the-art MrBayes, it is essential to re-emphasize two significant issues with VBPI-GNN:
>
> 1. The algorithm requires a pre-generated high-quality tree set to constrain the action space for tree construction. The performance of VBPI-GNN is highly contingent on the quality of this pre-generated tree set.
>
> 2. Due to the limitations on the action space imposed by the pre-generated tree set, VBPI-GNN cannot model the entire phylogenetic tree space. In other words, VBPI-GNN does not support the majority of phylogenetic trees. We are happy to offer a more detailed explanation of how VBPI-GNN constrains its model space. (In case there is confusion, this issue stands apart from our analysis in Table 2, Figure 2, and Figure S1, which highlight VBPI-GNN's limitations in learning suboptimal trees.)
>
> Prior to August 2023,  VaiPhy was the leading VI algorithm capable of modeling the entire tree space. However, as indicated in Table 1 (column $\phi$-CSMC), its  MLL estimation is significantly inferior to the state-of-the-art. This observation serves as a key motivation for our work: the design of a phylogenetic inference algorithm leveraging GFlowNets, with the objective of modeling the entire tree space and improving MLL estimation.
>
> Please also see the response to all reviewers, where we discuss the importance of modeling the entirety of the full phylogenetic tree space and VBPI-GNN's inferiority to PhyloGFN in this regard.
>
> ## Experiment specification in main text
>
> The $\pm$ are standard deviation and they are obtained from a single PhyloGFN model estimating the marginal log likelihood 10 times, using 1000 trajectories in each round. We are currently repeating the experiments for several runs to ensure significance. As described in the response to all reviewers, we are also repeating 3 sets of experiments with reduced training examples. We will update the MLL results table with new runs once we have obtained all the results and detail the setup in the text.
>
> ## Explanation of tree representation is unclear
> Perhaps it was not made clear how the node representations are derived: in Section 3.1, we describe Felsenstein’s algorithm and Fitch’s algorithm, which compute vectors of features for each node. For a given subtree with root node $r$, these features are exactly the vectors computed by these two dynamic programming algorithms for node $r$ (used for full Bayesian and parsimony-based inference, respectively).
>
> Section 3.1 describes in detail how these two features can be calculated recursively using the features of the child nodes.  We will update the manuscript to make the reference more clear in the tree representation section.
>
> ## Possible misunderstanding regarding Table 2 and Figure 2
>
> The analysis presented in Table 2 and Figure 2 is not intended to show diversity. Recall that we have obtained similar MLL compared to VBPI-GNN (e.g., -7018.6 for VBPI-GNN and -7018.9 for PhyloGFN on DS1). However, MLL mostly measures how well the algorithms model the high-probability trees. Therefore, in Table 2 and Figure 2, we are trying to answer the question of how well the algorithms model the rest of the tree space.
>
> In Table 2, Figure 2, and Figure S1, we show that on PhyloGFN is better at modeling the entire tree space compared to VBPI-GNN. For a given set of trees, we compare the model-learned sampling probability against their ground truth posterior density. For each data set, we performed the analysis on three sets of data with high/medium/low posterior densities. The analysis shows that while both algorithms model high-probability trees well, PhyloGFN is much better at modeling the rest of tree space.
>
> Please also see the responses to all reviewers regarding the importance of modeling the full space.

---

> > ### Author Response · Authors · 2023-11-17
> >
> > ## Metric to show diversity
> > We do address the diversity of samples with Temperature-conditioned PhyloGFN. Recall that PhyloGFN-Parsimony learns to fit an energy distribution of tree topologies where the parsimony scores of trees are scaled by the temperature $T$.  At a small $T$, trees with low parsimony score would dominate the energy distribution, so the sample diversity would be low. For example, in dataset DS1, there is only 1 tree topology with the minimum parsimony score, and it would make up most of the sample as $T\to0$.
> >
> > Therefore, we have proposed the temperature-conditioned PhyloGFN, a single sampler that accepts the temperature $T$ as input and is able to sample from the target energy distribution at that temperature. Figure 3 on page 9 shows the sampling results of our temperature-conditioned PhyloGFN on DS1. On the left, we plot the histograms of parsimony scores of trees sampled at different $T$. We see that as we increase $T$ gradually from 1 to 8, the sample diversity increases accordingly. The plot on the right shows that under different temperatures, PhyloGFN-Parsimony is still able to sample proportionally from the ground truth distribution.
> >
> > ## Validate PhyloGFN against ground truth evolution data
> >
> > For any given set of species, their ground truth evolutionary history and relationships are generally unknown, so we cannot directly evaluate the accuracy of any of the phylogenetic inference approaches. When this is of interest, researchers use simulated data, where the tree is known, to assess the ability of an approach to recover the correct tree. However our objective is slightly different: our goal is to sample from the posterior distribution over trees defined by a given set of sequences. This is a purely mathematical problem; the correctness of the sampling produced by a given method can thus be assessed without the knowledge of the evolutionarily correct tree. This is what is done in our paper and is consistent with a long line of prior work in phylogenetic inference.
> >
> > We hope that our responses have addressed your questions and concerns sufficiently. Please let us know if we can provide any more clarifications.

---

> > > ### Author Response · Authors · 2023-11-17
> > >
> > > ## Possible misunderstandings regarding parsimony analysis vs. Bayesian inference
> > > Parsimony score and marginal likelihood are not two metrics to assess the same algorithm. Phylogenetic inference is the task of inferring evolutionary relationships among species. Bayesian inference and parsimony-based inference are two different types of methods/approaches to perform phylogenetic inference:
> > >
> > > - Given a pre-defined probabilistic evolution model, Bayesian inference algorithms learn a model to fit the posterior distribution of trees.
> > > - Parsimony-based methods solve a discrete optimization problem: finding tree topologies with minimum parsimony scores.
> > >
> > > These are two separate types of problems that usually are solved with two different families of algorithms. While our proposed PhyloGFN is a robust framework that can solve both types of problems, PhyloGFN-Bayesian and PhyloGFN-Parsimony are still two separate algorithms. MLL is used to evaluate how well PhyloGFN-Bayesian fits the posterior distribution. The minimum parsimony score of sampled trees is used to evaluate if PhyloGFN-Parsimony can solve the minimization problem.

---

> ### Author Response · Authors · 2023-11-21
>
> Dear Reviewer SMRb,
>
> We have responded to your questions and comments above. Could you please let us know if you have any further questions before the end of the rebuttal period and if they have affected your assessment of the paper?
>
> We have also just posted a revised pdf. Please see the comment to all reviewers for details of the changes.
>
> Thank you,
>
> The authors

---

> ### Author Response · Authors · 2023-11-22
>
> Dear Reviewer SMRb,
>
> We have completed the repeated experiments for all datasets and we have updated the manuscript. We believed we have also addressed your questions and concerns in our previous responses.
>
> Could you please let us know if you have any further questions before the end of the rebuttal period and if they have affected your assessment of the paper?
>
> Thank you,
>
> The authors

---

### Official Review · Reviewer_iXKc · 2023-10-31

**Soundness:** 3 good
**Presentation:** 4 excellent
**Contribution:** 3 good
**Rating:** 8
**Confidence:** 4

**Summary:**

This paper tackles the related problems of Bayesian phylogenetic inference and maximum parsimony phylogenetic inference.  These problems have recently seen a lot of development in the machine learning space with a flurry of variational approaches.  The novelty in this paper is framing the problem as a Markov Decision Problem, where one needs to sequentially build a tree from the bottom up by joining the roots of rooted trees (starting with each leaf being its own rooted tree with one node).  This paper tries to learn a good policy for the MDP (i.e., a generative model for building trees) using the recently developed GFlowNets.  The paper puts some effort into finding provably good features to use when learning the policy, and showing that the optimal policy would, in fact, result in a distribution over trees equivalent to the posterior.  The authors apply their approach to a number of standard benchmarking datasets and find reasonable performance, particularly for low probability tree topologies.

**Strengths:**

* The paper is clear and well-written.
* The approach is interesting, conceptually simple, and provides a nice, efficient way to generate distributions over trees that support all of tree space (as opposed to relying on a pre-defined subset of tree space like VBPI).
* The theoretical results and connections to Felsenstein's and Fitch's algorithms are really nice.
* The performance across the full space of trees is promising.

**Weaknesses:**

* I know that it is common in the Bayesian Phylogenetics field, but I am uncomfortable with using the Marginal log-likelihood (MLL) as a measure of the accuracy of the posterior.  As the authors note, taking $K \to \infty$ in equation (5) or the log of (S1)results in the true MLL regardless of the distribution used.  For finite $K$, both the bias and variance of the estimated MLL will depend on the learned posterior and it is incredibly difficult for me to compare methods.  For example in Table 1, on DS7, GeoPhy is bolded for having the smallest mean MLL, but its standard error is huge, which suggests to me that by some measures it may not have learned a very good posterior.  Is there a different task that could get at the quality of the posterior in a more interpretable way?
* I also find it somewhat surprising how much VBPI-GNN outperforms PhyloGFN on the MLL estimation task.
* Saying that VBPI-GNN is "severely" limited in its applicability to postulating alternative phylogenetic theories feels like far too strong of a statement, especially in light of Table 2 -- for trees with non-negligible posterior probability VBPI-GNN is quite accurate.  I agree that VBPI being unable to put mass on all of tree space is conceptually displeasing, but I don't think the presented evidence supports the claim that VBPI would result in any real-world failures of inference.
* Are the axes flipped on Figure 2? If I understand correctly the x-axis should be the unnormalized posterior under the long run of MrBayes, and the y-axis is the unnormalized posterior for either GFN or VBPI.  Why do the points fall on different x-axis ranges for the two columns but very similar y-axis ranges?  (Similarly for Figure S1)
* It would be nice to include some information about runtime and to include some training curves.  Are these models difficult to train?  How sensitive is training to the choice of distribution over $\tau$?  Etc...

Typos:
* "Given a set of observed sequence" --> "Given a set of observed sequences"
* "Each action chooses a pair of trees and join them" --> "Each action chooses a pair of trees and joins them"
* Is it important or a typo in Proposition 1, Lemma 1, and Lemma 2 that only the branch lengths of the first tree are different?  I.e., it is $(z_1, b_1)$ and $(z_1', b_1')$ but then $(z_2, b_2)$ and $(z_2', b_2)$ and so on.  I don't see that used anywhere.
* In the statement of Lemma 1 I believe it should be $(b(e_{uv}), b(e_{uw}))$ not $(b(e_{uv}), (b(e_{uv}))$.
* This sentence needs substantial rewording: "Note that $R(x) \ne R(x') even $s_1$ and $s_2$ share the same Fitch feature because two trees can have different parsimony score when their root level Fitch feature equals."
* In the proof of proposition 2, I believe that one must multiply by $\frac{\exp \frac{\sum_i M(z_i)}{T}}{\exp \frac{\sum_i M(z_i)}{T}}$, not $\frac{\frac{\sum_i M(z_i)}{T}}{\frac{\sum_i M(z_i)}{T}}$.
* At the bottom of the first paragraph on p. 17 should it be $-\log P(\mathbf{Y} | z,b) P(b)$ not $-\log P(\mathbf{Y} | z,b) P(z)$ in order to match what is in the reward function?
* "ground-trueth" --> "ground-truth"

**Questions:**

* I am being a bit of a devil's advocate here, and it is a minor point, but, who cares about extremely unlikely trees?  Difference in log-likelihood on Figure 2 suggests that the trees with lowest posterior support are about $10^{-66}$ less likely than the trees with highest posterior support.  Does saying that those trees have zero probability really matter? Is there a real world use-case where knowing the posterior probability of those trees more accurately would be useful?
* Would it be possible to use the machinery presented in this paper just for topologies in the Bayesian setting by somehow marginalizing out branch lengths at each step?  This is obviously not necessary for the present manuscript, I am just curious.
* A preprint that was posted shortly after the ICLR deadline is highly relevant: https://arxiv.org/abs/2310.09553.  Since that was posted after the ICLR deadline, it did not influence my review of this paper.  The method in that preprint, ARTree, is based on reinforcement learning and also seeks to learn a posterior via framing a tree as the outcome of sequential tree building.  Yet, the training objectives and MDP formulation are substantially different, and so I do not see ARTree as reducing in any way the novelty of the present work.  I just bring this up in the hopes that it is interesting/helpful to the authors.

---

> ### Author Response · Authors · 2023-11-17
> **Response for reviewer iXKc**
>
> Thank you for feedback. We answer your questions and concerns below.
>
> ## On axes in Figure 2
>
> In Table 2 and Figure 2, we compute log sampling probability of PhyloGFN and VBPI-GNN and compare them to the ground-truth unnormalized posterior density for optimal trees and suboptimal trees. Log sampling probability (X-axis) is calculated using the algorithms in comparison (PhyloGFN and VBPI-GNN). Ground-truth unnormalized density (Y-axis) of tree $(z,b)$ is directly calculated as $P(Y,z,b) = P(Y|z,b)P(z)P(b)$. We assume a uniform distribution over tree space, hence $P(z) = \frac{1}{(2n-5)!!}$  and $P(b) = \prod_e P(b(e)), \ \ b(e) \sim \text{Exp}(10)$.
>
> For the reviewer’s question "why do points fall in similar Y ranges but different X ranges", we see two interpretations.
>
> If we interpret it as "comparing the VBPI-GNN plot to the corresponding PhyloGFN plot, why do the points fall in similar Y ranges but different X ranges": to ensure fair comparison, we are using the same set of trees. As the Y-axis corresponds to the ground truth joint density, they indeed should have the same range. The X-axis labels the sampling probability learned by the two models. In the VBPI-GNN plot, the points falling in a narrower X range indicate that VBPI-GNN's sampling probabilities for these suboptimal trees are learned incorrectly: VBPI-GNN has overestimated the posterior probability of these suboptimal trees.
>
> If we interpret it as "why do the X-axis and Y-axis ranges within each plot differ", this is because the Y-axis shows the joint log-density $\log P(Y,z,b)$, not the posterior log-density $\log P(z,b|Y)$. For a perfectly trained model, all points in the graph would lie on a diagonal line with slope 1 and intercept equal to the MLL $\log P(Y,z,b)-\log P(z,b|Y)=\log P(Y)$.
>
> ## Interpretable analysis for Bayesian inference algorithms
>
> We agree with the reviewer’s comment. Using the MLL alone, it can be difficult to assess and interpret the relative performance of different algorithms, particularly when the estimated MLLs are similar or have high variance. This is precisely what motivates our quantitative and qualitative analysis that compares the learned sampling probability of PhyloGFN and VBPI-GNN to the ground truth unnormalized posterior density (Table 2, Figure 2, and Figure S1). These analyses show that:
>
> 1. For all datasets except DS7, although PhyloGFN has slightly lower MLL than VBPI-GNN, both algorithms are able to model the high-probability trees well. Therefore, from a practical standpoint, PhyloGFN is at least as good as VBPI-GNN in its tree sampling capability.
>
> 2. For suboptimal trees, both the correlation scores (Table 2) and scatter plots (Figure 2, Figure S1) show that PhyloGFN is actually superior to VBPI-GNN at modeling suboptimal trees.
>
> ## PhyloGFN'S MLL estimation is inferior to VBPI-GNN's
> We have identified three potential reasons for VBPI-GNN's better MLL estimation:
>
> 1. PhyloGFN models branch lengths using discrete multinomial distributions. When estimating MLL, the branch lengths model is converted into a piecewise-constant continuous form, inducing a small quantization error. In contrast, VBPI-GNN models branch lengths with a continuous distribution directly.
>
> 2. VBPI-GNN utilizes a set of pre-generated high-likelihood trees to restrict the search space. Since most trees with high posterior density also have high likelihood, it is easier for VBPI-GNN to explore the high posterior density region. **(In this sense, the comparison is not entirely fair, since VBPI-GNN uses information that the training of PhyloGFN does not have access to.)**
>
> 3. There is room for improvement in model fitting for DS7. Both the MLL study and the study on sampling probability against ground truth indicate that PhyloGFN poorly models the high-probability trees for DS7.
>
> In future work, we intend to address these limitations and improve PhyloGFN's MLL estimation.
>
> ## Significance of modeling the entire tree space well
> Please see the response to all reviewers, where we describe important biological questions where properly sampling suboptimal trees is important.

---

> > ### Author Response · Authors · 2023-11-17
> >
> > ## Additional training information
> > For the running time and hardware resources comparison, please refer to our the answer to all reviewers. We will update the manuscript to include more training related details and training curves.
> >
> > To address your two questions:
> >
> > 1. **Are models difficult to train?** PhyloGFN trains smoothly on all datasets except DS7. We observe that temperature annealing plays a significant role in exploring the modes of the reward landscape. During cascading temperature drops, a notable spike and subsequent reduction in training loss occurs each time the temperature is reduced. (On DS7, some instability occurs at one of the temperature drops and the loss then remains at a persistently high value.)
> >
> > 2. **How sensitive is training to the choice of distribution over $\tau$?** If we understand correctly, you are referring to the choice of training policy. To promote mode discovery, we use a replay buffer, off-policy exploration via random actions, and temperature annealing, as described in our text. We did not experiment extensively with these settings but can add a comparison to pure on-policy training for the final version.
> >
> > ## Inferring tree topologies in the Bayesian setting
> > Indeed, this is an interesting question. Although, as far as we know, exact marginalization over lengths at each step would be intractable, a trained PhyloGFN model that samples both tree topologies and branch length can be used to sample only topologies, by simply "forgetting" the branch lengths. If the PhyloGFN is perfectly trained, the distribution of topology samples obtained in this way is the true posterior over tree topologies. The marginal likelihood of a topology could then be estimated via importance weighting, similar to what is done in our MLL evaluation.
> >
> > ## ARTree
> > Thank you for sharing this paper with us. At the inception of the project, we had considered constructing phylogenetic trees in a similar top-down fashion, but in the end we designed the current bottom-up PhyloGFN, because this way we can obtain a more efficient subtree representation based on the Fitch/Felsenstein algorithms.
> >
> > One property of the ARTree construction process is that one phylogenetic tree is mapped uniquely to one construction path. For this, the order of input sequence needs to be predetermined. However, if an MDP is designed exactly this way, actions at each step may be less intuitive. Depending on the order of sequences, the agent may have difficulty identifying the optimal decision for inserting the new species to the current partial tree topology. Of course, through learning from a large set of examples, the agent will know that its current action will be beneficial at certain points in the future. However, let us now consider a more challenging problem: learning a generalized agent that can take any input set of sequences and perform phylogenetic inference. Given an agent and a set of sequences that the model has never seen before,  it may have difficulty estimating each action’s future impact without seeing any examples in advance. On the other hand, with our bottom up approach, we believe it may be easier to solve the generalized problem. For example in parsimony analysis, the best action in many scenarios is to select and join the two subtrees with the closest root Fitch feature, which is a relatively easy concept to learn for PhyloGFN.
> >
> > ## Typos
> > Thank you for pointing out the typos in our manuscript. We have corrected them.
> >
> > We hope that we have satisfactorily addressed your questions and concerns. If there are further points that require clarification, please let us know.

---

> > > ### Comment · Reviewer_iXKc · 2023-11-18
> > >
> > > Thank you for the response and for the interesting paper.

---

> ### Author Response · Authors · 2023-11-21
>
> Dear Reviewer iXKc,
>
> Thank you for taking the effort of reviewing our paper.
>
> The authors

---

### Official Review · Reviewer_tiKV · 2023-10-31

**Soundness:** 3 good
**Presentation:** 2 fair
**Contribution:** 2 fair
**Rating:** 5
**Confidence:** 4

**Summary:**

The paper presents an application of GFlowNets for phylogenetic inference. The approach encompasses both Bayesian posterior inference and parsimony-based inference (MLE with the assumption of site independence). To address the continuous nature of branch lengths, they are transformed into discrete bins, making the phylogenetic inference problem discrete. GFlowNets are parameterized by neural networks, teaching them a policy to generate phylogenetic trees by selecting and connecting pairs of subtrees. With its probabilistic policy, the system can produce a distribution of phylogenetic trees suitable for posterior inference. In the MLE scenario, the model employs a temperature parameter that, when annealed, ensures the GFlowNet samples align with the MLE samples. The training of GFlowNets utilizes standard trajectory balance. The results suggest that the approach matches the performance of variational inference-based methods for Bayesian posterior inference, and in the MLE case, it is comparable to traditional greedy heuristic-based search algorithms.

**Strengths:**

- *Innovative Application*: Utilizing GFlowNets for phylogenetic inference is a novel idea, showcasing the versatility of GFlowNets in unique problem settings.
- *Competitive Performance*: The approach not only matches up to VBPI-GNN-based methods in Bayesian posterior inference but offers capabilities beyond them, like generating from arbitrarily filled-in subtrees which preceding methods could not tackle.
- *Estimating probability of suboptimal structures*: The ability to outperform VBPI methods in estimating probabilities of suboptimal structures is a notable accomplishment.

**Weaknesses:**

- *Performance vs. Efficiency*: While GFlowNets might perform comparably to PAUP* in the parsimony-based inference setting, the real differentiator would be computational efficiency on a new inference task. Unfortunately, no wall-clock time data is provided, making it challenging to discern any advantages of GFlowNets in this scenario.
- *Methodological Novelty*: The paper does not seem to bring forth significant machine learning methodological advancements, with much of the methodology being straightforward applications without any notable novel ML innovations for the particular problem at hand. The biggest methodological contribution is the use of discretized bins for branch lengths, which is a compromised solution.
- *Utility of Results*: While the GFlowNets surpass VBPI methods in estimating probabilities for suboptimal structures, the practical significance of this in the context of phylogenetic applications remains unexplained. The experiments focus solely on the accuracy of probability estimates and fail to provide insights into their relevance for the broader application.

**Questions:**

- *Time Efficiency*: Given the comparable performance of GFlowNets and PAUP*, can the authors provide information on the computational efficiency (wall-clock time) of GFlowNets to discern its advantages or disadvantages?
- *Utility of Suboptimal Structures*: Can the authors elucidate the practical implications of having estimates for suboptimal structures in the context of phylogenetics? Is there a tangible benefit in real-world applications?

---

> ### Author Response · Authors · 2023-11-17
> **Response for reviewer tiKV**
>
> Thank you for your feedback.  We hope the answers below clarify your concerns.
>
> ## Comparing running time with PAUP\*
>
> For parsimony analysis, our models are trained with 25.6 million examples on all datasets, with training duration of 3-8 days. The heuristic search algorithm of PAUP\*; is able to discover optimal trees in less than 30 CPU seconds for all datasets. This is largely due to the optimization efforts that have been put into the software over the past two decades.
>
> In this context, it is interesting to compare the number of trees evaluated by each method. For example in DS8, PAUP*; has attempted a total of 8,113,732 tree shape rearrangements (TBR branch swapping) for finding the 21 most parsimonious trees, while PhyloGFN is trained with 25,600,000 trajectories.
>
> It is worth bearing in mind that, in contrast to PAUP*, (1) **Bayesian phylogenetic inference algorithms, such as PhyloGFN, tackle the far more challenging problem** of sampling from a distribution over the entire phylogenetic tree space where the most parsimonious trees are the global minima, and (2) the amortization technique allows us to control the distribution of tree samples using a single model conditioned on the temperature variable.
>
> Despite all models undergoing training with 25.6 million trajectories, they exhibit the capability to produce parsimonious trees at significantly earlier stages in the training process. At intervals of every 5\% of the training process for each model, we generate 10,000 samples and evaluate the parsimony scores of the sampled trees. This table shows the percentage of training time required for each model to sample at least one of the most-parsimonious trees:
>
> | DS1 | DS2 | DS3 | DS4 | DS5 | DS6 | DS7 | DS8 |
> |-----|-----|-----|-----|-----|-----|-----|-----|
> | 15\% | 20\% | 20\% | 45\% | 65\% | 75\% | 45\% | 45\% |
>
> We haven't fine-tuned the training setup specifically for fast parsimonious tree retrieval. By training with a more aggressive temperature annealing schedule, integrating an early stopping mechanism, and various code and hardware optimizations, it could become feasible to significantly reduce the number of training examples needed to generate the most parsimonious trees. However, we emphasize again that the goal of PhyloGFN and other Bayesian phylogenetic inference algorithms is to model the full distribution over trees, not simply to find the most parsimonious ones.
>
> For running time for Bayesian inference and comparison with other variational inference algorithms, please refer to the response to all reviewers.
>
> ## Utility of suboptimal structures
> This is a good question; please refer to our answer to all reviewers.

---

> > ### Author Response · Authors · 2023-11-17
> >
> > ## ML innovation
> > We acknowledge the reviewer's concern regarding the methodological contribution and novelty of PhyloGFN. We believe that our study is an innovative application of the GFlowNet framework to the problem of phylogenetic inference. Our approach outperforms two other state-of-the-art variational inference methods, VaiPhy and GeoPhy, both of which also sample from the entire phylogenetic tree space. The success of our method relies on several key methodological innovations:
> >
> > First, we have proposed a Markov decision process that allows us to generate phylogenetic trees in a bottom-up fashion, through iteratively joining pairs of subtrees and creating new roots, until a complete tree is formed. To the best of our knowledge, this is the first bottom-up phylogenetic tree construction scheme proposed in the ML-for-phylogenetics literature. In VBPI-GNN and its more recent extension ARTree, trees are constructed in a top-down fashion, which requires learning representations of the ancestral nodes using Dirichlet energy minimization and graph neural networks. For ARTree in particular, the Dirichlet energy minimization and graph representation learning at each step raises questions of scalability to larger trees.
> >
> > Second, our bottom-up tree construction scheme allows us to employ two kinds of provably-powerful and yet easy-to-compute features for the ancestral nodes -- those produced by applying the Fitch and Felsenstein algorithms. After joining the pair of subtrees, the newly created root node is represented by these features. This offers several advantages: (1) it allows us to directly model the new subtree without introducing any learnable parameters such as a graph neural network; (2) this feature representation is very fast to compute exactly; (3) it is scalable to larger phylogenetic inference datasets. Again, to the best of our knowledge, we are the first to design and employ these features in the machine learning context, and we theoretically justify that these features provide sufficient information for any optimal sampling policy.
> >
> > Third, the parsimony problem is traditionally treated as one of discrete optimization. In this work, we frame it as a hierarchical inference problem: we train a PhyloGFN to sample from an energy distribution over tree topologies, and by manipulating the temperature on which the model is conditioned we can control the trade-off between the sampled tree parsimony scores and diversity. At inference time, we can sample from a series of energy distributions at different temperatures.
> >
> > We hope we have addressed all of your concerns. Please let us know if something else needs further clarification or if we missed something.

---

> ### Author Response · Authors · 2023-11-21
>
> Dear Reviewer tiKV,
>
> We have responded to your questions and comments above. Could you please let us know if you have any further questions before the end of the rebuttal period and if they have affected your assessment of the paper?
>
> We have also just posted a revised pdf. Please see the comment to all reviewers for details of the changes.
>
> Thank you,
>
> The authors

---

> ### Comment · Reviewer_tiKV · 2023-11-21
>
> Thank you for your response. "PAUP*; is able to discover optimal trees in less than 30 CPU seconds for all datasets." It seems much faster than training a GFN. In this case, the best utility of a trained GFN might be zero-shot learning on a new inference task. Have you tried that and if so, how dos the test inference time compare with PAUP*?
>
> On modeling the distribution of suboptimal trees, if this is what PhyloGFN is better at compared with MCMC and VI, i.e. provide probability estimate or identify substructures, I really hope to see experiments that illustrate this in experiments. Especially given that this is a ML-application paper, I would hope to see how this proposed method unlock interesting applications that previous methods fail to achieve.

---

> > ### Author Response · Authors · 2023-11-21
> > **Response for reviewer tiKV**
> >
> > Dear reviewer tiKV,
> >
> > Thank you for your comments and questions. We address your specific questions and comments below.
> >
> > ## Applying PhyloGFN to zero-shot learning
> >
> > > the best utility of a trained GFN might be zero-shot learning on a new inference task. Have you tried that
> >
> > We agree with the reviewer that extending PhyloGFN to zero-short learning is a valuable direction, but as of currently we have not tested PhyloGFN on new inference tasks with unseen sequences. We must point out that zero-shot learning is a highly challenging task of phylogenetic inference which has yet to be explored in any existing machine learning literature, but we do intend to answer this question with our PhyloGFN in a follow-up work.
> >
> > In order to enable zero-shot learning, that is to apply a pre-trained PhyloGFN on any arbitrary set of sequences and inferring their phylogeny, further modifications on the model architecture and training is required. Just to give a few examples:
> > - as is mentioned in appendix section D, we currently apply a linear transformation on the input sequences to obtain lower-dimensional embeddings, before they are input to the Transformer encoder. However, this linear transformation is unable to process sequences of varying length. We are currently investigating more flexible models to model to embed sequences of arbitrary length.
> > - modification to PhyloGFN training is also needed. At the moment, each model is trained on only one dataset with up to 64 sequences. For the zero-shot learning, it would be imperative for our PhyloGFN to be trained on a large amount of sequences across many datasets, in order to ensure generalizability.
> >
> > Once again, we thank our reviewer for bringing up this research direction, and as part of our future plans, we have highlighted this direction in the conclusion of the manuscript.
> >
> > ## Possible confusions on advantage of PhyloGFN
> >
> > > The approach not only matches up to VBPI-GNN-based methods in Bayesian posterior inference but offers capabilities beyond them, like generating from arbitrarily filled-in subtrees which preceding methods could not tackle
> > >
> > > On modeling the distribution of suboptimal trees, if this is what PhyloGFN is better at compared with MCMC and VI, i.e. provide probability estimate or identify substructures
> >
> >
> > As in our response to Reviewer SMRb, We would like to re-emphasize two significant issues of VBPI-GNN:
> >
> > 1. VBPI-GNN requires a pre-generated high-quality tree set to constrain the action space for tree construction. The performance of VBPI-GNN is highly contingent on the quality of this pre-generated tree set.
> >
> > 2. Due to the limitations on the action space imposed by the pre-generated tree set, VBPI-GNN cannot model the entire phylogenetic tree space. In other words, VBPI-GNN does not support the majority of phylogenetic trees. We are happy to offer a more detailed explanation of how VBPI-GNN constrains its model space. (In case there is confusion, this issue stands apart from our analysis in Table 2, Figure 2, and Figure S1, which highlight VBPI-GNN's limitations in learning suboptimal trees.)
> >
> > Prior to August 2023, VaiPhy was the leading VI algorithm capable of modeling the entire tree space. However, as indicated in Table 1 (column $\phi$-CSMC), its MLL estimation is significantly inferior to the state of the art. This observation serves as a key motivation for our work: the design of a phylogenetic inference algorithm leveraging GFlowNets, with the objective of modeling the entire tree space and improving MLL estimation.
> >
> > We can summarize the advantages of PhyloGFN as follows:
> > - Comparing with VI algorithms that model the entire tree space, PhyloGFN improves significantly the performance in term of MLL estimation. Comparing with VBPI-GNN, the state-of-the-art VI algorithm, PhyloGFN models the entire tree space and achieves competitive MLL estimation without the need of a pre-generated high-quality tree set. Moreover, even if we consider limited modeling space of VBPI-GNN, PhyloGFN is superior at modeling sub-optimal trees.
> > - Comparing with MCMC, the advantages of PhyloGFN are (i). PhyloGFN is an amortized sampler, able to very quickly generate a large number of independent tree samples once it has been trained, (ii). scalability to high-dimensional distribution
> >
> > ## Comparability with PAUP*
> >
> > We want to reiterate the point from our original response that PAUP* is not a Bayesian phylogenetic inference algorithm, unlike PhyloGFN (and many other works in Bayesian phylogenetic inference). The goal of algorithms such as PhyloGFN is amortized sampling of the entire tree space, not just discovery of optimal trees. The more difficult problem and the task of amortization contribute to high running time for all amortized Bayesian phylogenetics algorithms, which take hours to converge (see the response to all reviewers).

---

> > > ### Author Response · Authors · 2023-11-21
> > >
> > > ## Merit of PhyloGFN as an ML-application work
> > >
> > > > Especially given that this is a ML-application paper, I would hope to see how this proposed method unlock interesting applications that previous methods fail to achieve
> > >
> > > Effectively modeling the entirety of the tree space poses a significant challenge, and it stands as the primary issue that researchers in this field are currently addressing. Among this year's NeurIPS accepted papers, two notable contributions (GeoPhy and ARTree) present innovative approaches to model the entire tree space while maintaining high MLL estimation performance. The previous response has already highlighted the machine learning contributions in this work, namely, the unique bottom-up tree construction process and the application of a diversity-seeking reinforcement learning method (GFlowNet )framework to the problem. Our proposed algorithm PhyloGFN models the entire tree space and significantly improves MLL estimation compared to prior works that focus on this aspect.
> > >
> > > ## On suboptimal trees
> > > We appreciate the reviewer’s suggestion that there could be further interesting experiments concerning the use of PhyloGFN to analyze suboptimal phylogenetic trees. We note that in our response to all reviewers we have discussed some of these application areas where PhyloGFN could be of high interest. For example, understanding cancer and virus evolution where the capability of modeling the full tree space, identifying common substructures of trees and quantifying uncertainties of phylogenetic hypotheses are particular advantages of our PhyloGFN. While these are exciting directions for future work, unfortunately we would not be able to include them during the rebuttal period due to time constraints.  We respectfully note that the breadth and thoroughness of our experimentation is quite in line with related works exploring novel methods for phylogenetic inference, such as VBPI-GNN, VaiPhy and the previously mentioned concurrent work (GeoPhy and ARTree).
> > >
> > > We thank our reviewer for bringing up these research directions. We hope that our responses have addressed your questions and concerns sufficiently. Please let us know if we can provide any more clarifications.

---

> ### Author Response · Authors · 2023-11-22
>
> Dear Reviewer tiKV,
>
> We have updated the manuscript to add more references delineating the importance of sampling suboptimal trees (section appendix H). We thank you again for suggesting interesting future work directions. We believed we have addressed your questions and concerns in our previous responses.
>
> Could you please let us know if you have any further questions before the end of the rebuttal period and if they have affected your assessment of the paper?
>
> Thank you,
>
> The authors

---

> ### Comment · Reviewer_tiKV · 2023-11-22
>
> Thank you for the detailed response. I fully agree with the importance of sampling suboptimal trees and believe this is what the model should focus on. I also agree that current experiments show that GFN is promising in sampling from the tree distribution. But just that I wish to see something more interesting that can be achieved with GFN (in the tree sampling setting) beyond log-likelihood since it is an application paper. The time comparison with MCMC is helpful in understanding how GFN trains. I don't have further questions at the moment. My current standpoint is neural towards accepting so I will not change my score, although 5.5 better reflects where I stand.

---

> ### Author Response · Authors · 2023-11-23
>
> Dear Reviewer tiKV,
>
> Thank you for your effort in reviewing our work and so thoroughly justifying your assessment. We respect your decision.
>
> Nevertheless, we still want to re-emphasize the following, not to convince you to change the score, but to clear up any possible misunderstandings for you or other reviewers: while modeling suboptimal trees well is a highlight for PhyloGFN, it is only one consequence of our powerful approach to Bayesian inference via sequential construction, rather than the main focus. To elaborate:
>
> - Our improved modeling of the entire tree space without the need for pre-generated trees is an important contribution. Modeling the entire tree space is not quite the same as "modeling well suboptimal trees" or "generating from arbitrarily filled-in subtrees" as you described in your original comment. Without prior knowledge (for example, pre-generated trees), the algorithm would have no idea what trees are suboptimal or arbitrary unless it can properly explore and model the tree space.
>
> - Modeling the full tree space faithfully is perhaps the **primary** issue that researchers in Bayesian phylogenetics are currently addressing: the technical challenges intensify across multiple scales when the algorithm does not depend on pre-generated tree sets to constrain the model space. Past and concurrent works in the last two years (VaiPhy, GeoPhy, and the concurrent ARTree) have proposed innovative solutions to tackle these challenges, and our current work builds and improves upon this line of research, showing for the first time that deep reinforcement learning approaches bring unique advantages to this task.
>
> Thank you for your consideration.
>
> The authors

---

### Author Response · Authors · 2023-11-17

# Responses for all reviewers
We would like to thank all reviewers for their detailed comments on our paper. In this comment, we answer two concerns that have been raised by multiple reviewers.

##  Running time and hardware
PhyloGFN is trained on virtual machines equipped with 10 CPU cores and 10GB RAM for all datasets. We use one V100 GPU for datasets DS1-DS6 and one A100 GPU for DS7-DS8, although the choice of hardware is not essential for running our training algorithms.

The models used for the MLL estimation in Table 1 of the paper are trained on a total of 32 million examples, with a training wall time ranging from 3 to 8 days across the eight datasets. However, our algorithm demonstrates the capacity to achieve similar performance levels with significantly reduced training data. The table below presents the performance of PhyloGFN with 32 million training examples (**PhyloGFN Full**) and includes three repetitions of training with only 40\% of the training trajecories (**PhyloGFN Short**). Please note that the table is currently incomplete as some experiments are still in progress and will be updated later in the rebuttal period. Based on the results collected thus far, the shorter runs exhibit comparable performance to our full run experiments, and all conclude within 3 days.

| Experiment         | Run | DS1              | DS2              | DS3              | DS4              | DS5              | DS6              | DS7              | DS8              |
|--------------------|--------|------------------|------------------|------------------|------------------|------------------|------------------|------------------|------------------|
| PhyloGFN Full      | 1      | -7108.95 (0.06) | -26368.9 (0.28)  | -33735.6 (0.35)  | -13331.83 (0.19) | -8215.15 (0.2)  | -6730.68 (0.54)  | -37359.96 (1.14)    | -8654.76 (0.19) |
| PhyloGFN Full      | 2      | -7108.97 (0.05) | -26368.77 (0.43) | -33735.60 (0.40)| -13331.80 (0.31) | -8214.92 (0.27)    | -6730.72 (0.26) | -37360.59 (1.62)  | -8654.67 (0.39)  |
| PhyloGFN Full      | 3      | -7108.94 (0.05) | -26368.89 (0.29) | -33735.68 (0.46)| -13331.43 (0.84) | -8214.85 (0.28)    | -6730.89 (0.22) |  -37361.51 (2.89)       | -8654.86 (0.46)               |
| PhyloGFN Short     | 1      | -7108.97 (0.14) | -26368.9 (0.39)  | -33735.9 (0.91)  | -13332.04 (0.57) | -8215.38 (0.27) | -6731.35 (0.31) | -37362.03 (5.20)   |  -8655.8 (0.95) |
| PhyloGFN Short     | 2      | -7108.94 (0.22) | -26369.03 (0.31) | -33735.83 (0.62)| -13331.87 (0.31) | -8215.37 (0.26) | -6731.2 (0.40)   | -37363.43 (2.21)     | -8655.65 (0.37)        |
| PhyloGFN Short     | 3      | -7109.04 (0.08) | -26368.88 (0.32) | -33735.76 (0.75)| -13331.78 (0.37) | -8215.38 (0.25) | -6731.1 (0.38)  | -37362.37 (2.65)       | -8654.96 (0.46)       |
| PhyloGFN Short Running Time|                  | 20h40min         | 28h              | 35h40min         | 44h30m          | 51h40m          | 53h10min     | 60h20min         | 61h40           |
| PhyloGFN Long Running Time |                  | 62h40min         | 69h16min         | 80h20min         | 103h54min       | 127h50min       | 135h10min    | 174h3min         | 190h25min       |


## Runtime comparison with VI algorithms
We compare the running time of PhyloGFN with VI baselines (VBPI-GNN, Vaiphy, and GeoPhy) using the DS1 dataset. VBPI-GNN and GeoPhy are trained using the same virtual machine configuration as PhyloGFN (10 cores, 10GB ram, 1xV100 GPU). The training setup for both algorithms mirrors the one that yielded the best performance as documented in their respective papers. As for VaiPhy, we employed the recorded running time from the paper on a machine with 16 CPU cores and 96GB RAM.

For PhyloGFN, we calculate the running time of the full training process (PhyloGFN-Full) and four shorter experiments with 40\%, 24\%, 16\% and 8\% training examples.

The table below documents both the running time and MLL estimation for each experiment. While our most comprehensive experiment, PhyloGFN Full, takes the longest time to train, our shorter runs -- all of which conclude training within a day -- show only a marginal degradation in performance. Remarkably, even our shortest run, PhyloGFN - 8\%, outperforms both GeoPhy and $\phi$-CSMC, achieving this superior performance with approximately 60\% training time of GeoPhy.


|                | VBPI-GNN   | GeoPhy   | $\phi$-CSMC | PhyloGFN Full | PhyloGFN - 40\% | PhyloGFN - 24\% | PhyloGFN - 16\% | PhyloGFN - 8\% |
|----------------|------------|----------|-------------|------------------|----------------|-----------------|-----------------|---------------|
| Running Time   | 16h10min   | 8h10min | ~ 2h        | 62h40min         | 20h40min        | 15h40min        | 10h50min        | 5h10min       |
| MLL            | -7108.41 (0.14) | -7111.55 (0.07) | -7290.36 (7.23)  | -7108.97 (0.05) | -7108.97 (0.14) | -7109.01 (0.15) | -7109.15 (0.23) | -7110.65 (0.39) |

---

> ### Author Response · Authors · 2023-11-17
>
> ## Significance of modeling suboptimal trees well
>
> Reviewers tiKV and iXKc asked about the significance of modeling suboptimal trees well. Several approaches that rely on phylogenetic inference require not only the identification of optimal (maximum likelihood or most parsimonious) trees, but also the proper sampling of suitably weighted suboptimal trees. In evolutionary studies, this includes the computation of branch length support (i.e., the probability that a given subtree is present in the true tree), as well as the estimation of confidence intervals for the timing of specific evolutionary events. These are particularly important in the very common situations where the data only weakly define the posterior on tree topologies (e.g. small amount of data per species, or high degree of divergence between sequences). In such cases, because suboptimal trees vastly outnumber optimal trees, they can contribute to a non-negligible extent to the probability mass of specific marginal probabilities. Full modeling of posterior distributions on trees is also critical in studying tumor or virus evolution within a patient, e.g., to properly assess the probability of the ordered sequence of driver or passenger mutations that may have led to metastasis or drug resistance.

---

### Author Response · Authors · 2023-11-21

Dear Reviewers,

We have just uploaded a revised manuscript with improvements guided by your suggestions, as detailed in the individual responses. The changes are highlighted in red in the new pdf. Notable changes include:
- Mentioning that Felsenstein's algorithm is an instance of dynamic programming; explicitly defining the Felsenstein feature and Fitch features in Sections 3.1 and 3.2, explicitly referring to those features in Sections 4.1 and 4.2.
- Adding to Section 5.1 some details on MLL estimation and PhyloGFN's ability to model the full tree space, with pointers to the relevant Appendix sections for additional results, training details, and running time comparison.
- Adding four new sections to the appendix:
  - Appendix F: on resource requirements and running time comparison, as well as measurements of variance between runs.
  - Appendix G: an ablation study on  (1) bin size + bin number and (2) exploration policies.
  - Appendix H: discussion of the importance of modeling the distribution over suboptimal trees well.
  - Appendix L: training curves illustrating the effect of cascading temperature drops.

We also updated the table of repeated experiments.

Thank you again for your effort in reviewing our paper.

The authors

---

### Meta-Review · Program_Chairs · 2024-01-16

**Metareview:**

Note: The meta-review is written by the PCs.

This paper proposes a Generative FlowNet based approach for phylogenetic inference and shows competitive performance to existing MCMC and VI methods. In my opinion, the problem is relevant and challenging and progress on this will benefit the community. Methods more or less rely on existing methodologies but some new tools were employed to tackle the problem in hand. All reviewers like the main ideas of the paper with two reviewers giving a score of 8, while the other two giving a 6 and 5 respectively. The low-score reviewers engaged with the authors. Many of their issues were clarified but many big-picture concerns remains. Despite those, the reviewers were enthusiastic towards accepting this paper.

I agree with the reviewers' enthusiasm and recommend to accept the paper. Reviewers have given many suggestions to improve the paper, and many other points were highlighted during discussion. I hope that the authors will take this into account to improve the camera ready version.

**Justification For Why Not Higher Score:**

N/A

**Justification For Why Not Lower Score:**

The problem is relevant and challenging, while the solutions are new and useful.

---

### Decision · Program_Chairs · 2024-01-16

Accept (poster)